# What the success of brain imaging implies about the neural code

**Olivia Guest[1]\*, Bradley C Love[1,2]\***

[1]Experimental Psychology, University College London, London, United Kingdom; [2]The Alan Turing Institute, London, United Kingdom

**Abstract** The success of fMRI places constraints on the nature of the neural code. The fact that researchers can infer similarities between neural representations, despite fMRI's limitations, implies that certain neural coding schemes are more likely than others. For fMRI to succeed given its low temporal and spatial resolution, the neural code must be smooth at the voxel and functional level such that similar stimuli engender similar internal representations. Through proof and simulation, we determine which coding schemes are plausible given both fMRI's successes and its limitations in measuring neural activity. Deep neural network approaches, which have been forwarded as computational accounts of the ventral stream, are consistent with the success of fMRI, though functional smoothness breaks down in the later network layers. These results have implications for the nature of the neural code and ventral stream, as well as what can be successfully investigated with fMRI.

\*For correspondence: o.guest@ucl.ac.uk (OG); b.love@ucl.ac.uk (BCL)

**Competing interests:** The authors declare that no competing interests exist.

## Introduction

Neuroimaging and especially functional magnetic resonance imaging (fMRI) has come a long way since the first experiments in the early 1990s. These impressive findings are curious in light of fMRI's limitations. The blood-oxygen-level-dependent (BOLD) response measured by fMRI is a noisy and indirect measure of neural activity (*Logothetis, 2002*, *2008*; *O'Herron et al., 2016*) from which researchers try to infer neural function.

The BOLD response trails neural activity by 2 s, peaks at 5 to 6 s, and returns to baseline around 10 s, whereas neural activity occurs on the order of milliseconds and can be brief (*Huettel et al., 2009*). In terms of spatial resolution, the BOLD response may spill over millimeters away from neural activity due to contributions from venous signals (*Turner, 2002*). Likewise, differences in BOLD response can arise from incidental differences in the vascular properties of brain regions (*Ances et al., 2009*). Such sources of noise can potentially imply differences in neural activity in regions where there should not be.

The data acquisition process itself places limits on fMRI measurement. Motion artefacts (e.g., head movements by human subjects) and non-uniformity in the magnetic field reduce data quality. In analysis, three-dimensional images are constructed from slices acquired at slightly different times. Once collected, fMRI data are typically smoothed during analyses (*Carp, 2012*). All these factors place limits on what fMRI can measure.

Despite these weaknesses, fMRI has proved to be an incredibly useful tool. For example, we now know that basic cognitive processes involved in language (*Binder et al., 1997*) and working memory (*Pessoa et al., 2002*) are distributed throughout the cortex. Such findings challenged notions that cognitive faculties are in a one-to-one correspondence with brain regions.

Advances in data analysis have increased what can be inferred by fMRI (*De Martino et al., 2008*). One of these advances is multivariate pattern analysis (MVPA), which decodes a pattern of neural activity in order to assess the information contained within (*Cox and Savoy, 2003*). Rather

**eLife digest** We can appreciate that a cat is more similar to a dog than to a truck. The combined activity of millions of neurons in the brain somehow captures these everyday similarities, and this activity can be measured using imaging techniques such as functional magnetic resonance imaging (fMRI). However, fMRI scanners are not particularly precise – they average together the responses of many thousands of neurons over several seconds, which provides a blurry snapshot of brain activity. Nevertheless, the pattern of activity measured when viewing a photograph of a cat is more similar to that seen when viewing a picture of a dog than a picture of a truck. This tells us a lot about how the brain codes information, as only certain coding methods would allow fMRI to capture these similarities given the technique's limitations.

There are many different models that attempt to describe how the brain codes similarity relations. Some models use the principle of neural networks, in which neurons can be considered as arranged into interconnected layers. In such models, neurons transmit information from one layer to the next.

By investigating which models are consistent with fMRI's ability to capture similarity relations, Guest and Love have found that certain neural network models are plausible accounts of how the brain represents and processes information. These models include the deep learning networks that contain many layers of neurons and are popularly used in artificial intelligence. Other modeling approaches do not account for the ability of fMRI to capture similarity relations.

As neural networks become deeper with more layers, they should be less readily understood using fMRI: as the number of layers increases, the representations of objects with similarities (for example, cats and dogs) become more unrelated. One question that requires further investigation is whether this finding explains why certain parts of the brain are more difficult to image.

than computing univariate statistical contrasts, such as comparing overall BOLD activity for a region when a face or house stimulus is shown, MVPA takes voxel patterns into account.

Using MVPA, so-called 'mind reading' can be carried out — specific brain states can be decoded given fMRI activity (*Norman et al., 2006*), revealing cortical representation and organization in impressive detail. For example, using these analysis techniques paired with fMRI we can know whether a participant is being deceitful in a game (*Davatzikos et al., 2005*), and we can determine whether a participant is reading an ambiguous sentence as well as infer the semantic category of a word they are reading (*Mitchell et al., 2004*).

Representational similarity analysis (RSA), another multivariate technique, is particularly suited to examining representational structure (*Kriegeskorte et al., 2008*; *Kriegeskorte, 2009*). We will focus on RSA later in this contribution, so we will consider this technique in some detail. RSA directly compares the similarity (e.g., by using correlation) of brain activity arising from the presentation of different stimuli. For example, the neural activity arising from viewing a robin and sparrow may be more similar to each other than between a robin and a penguin.

These pairwise neural similarities can be compared to those predicted by a particular theoretical model to determine correspondences. For example, *Mack et al. (2013)* identified brain regions where the neural similarity structure corresponded to that of a cognitive model of human categorization, which was useful in inferring the function of various brain regions. The neural similarities themselves can be visualized by applying multidimensional scaling to further understand the properties of the space (*Davis et al., 2014*). RSA has been useful in a number of other endeavors, such as understanding the role of various brain areas in reinstating past experiences (*Tompary et al., 2016*; *Mack and Preston, 2016*).

Given fMRI's limitations in measuring neural activity, one might ask how it is possible for methods like RSA to be successful. The BOLD response is temporally and spatially imprecise, yet it appears that researchers can infer general properties of neural representations that link sensibly to stimulus and behavior. The neural code must have certain properties for this state of affairs to hold. What kinds of models or computations are consistent with the success of fMRI? If the brain is a computing

device, it would have to be of a particular type for fMRI to be useful given its limitations in measuring neural activity.

## Smoothness and the neural code

For fMRI to recover neural similarity spaces, the neural code must display certain properties. Firstly, the neural code cannot be so fine-grained that fMRI's temporal and spatial resolution limitations make it impossible to resolve representational differences. Second, a notion of *functional smoothness*, which we will introduce and define, must also be partially satisfied.

### Voxel inhomogeneity across space and time

The BOLD response summates neural activity over space and time, which places hard limits on what fMRI can measure. To make an analogy, $3 + 5$ and $6 + 2$ both equal $8$ through different routes. If different 'routes' of neural activity are consequential to the neural code and are summated in the BOLD response, then fMRI will be blind to representational differences.

To capture representational differences, voxel response must be inhomogeneous both between voxels and within a voxel across time. Consider the fMRI analogues shown in *Figure 1*; paralleling neurons with pixels and voxels with the squares on the superimposed grid. The top-left image depicts neural activity that smoothly varies such that the transitions from red to yellow occur in progressive increments. Summating within a square, i.e., a voxel, will not dramatically alter the high-level view of a smooth transition from red to yellow (bottom-left image). Voxel response is inhomogeneous, which would allow decoding by fMRI (cf. *Kamitani and Tong, 2005*; *Alink et al., 2013*). Altering the grid (i.e., voxel) size will not have a dramatic impact on the results as long as the square does not become so large as to subsume most of the pixels (i.e., neurons). This result is in line with basic concepts from information theory, such as the Nyquist-Shannon sampling theorem. The key is that the red and yellow pixels/neurons are topologically organized: their relationship to each other is for all intents and purposes invariant to the granularity of the squares/voxels (for more details see: *Chaimow et al., 2011*; *Freeman et al., 2011*; *Swisher et al., 2010*).

In contrast, the center-top image in *Figure 1* involves dramatic representational changes within voxel. Each voxel (square in the grid), in this case, will produce a homogenous orange color when its contents are summated. Thus, summating the contents of a voxel in this case obliterates the representational content: red and yellow; returning instead squares/voxels that are all the same uniform color: orange. This failure is due to sampling limits that could be addressed by smaller voxels (see

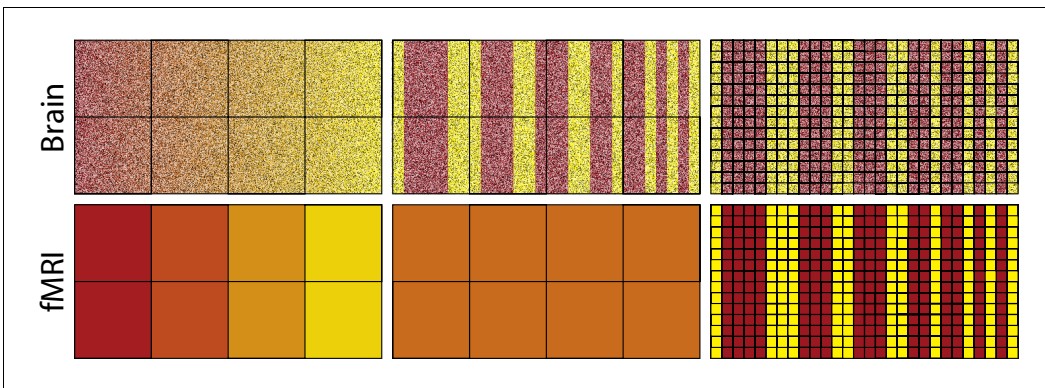

**Figure 1.** The activity of neurons in the top-left panel gradually changes from left to right, whereas changes are more abrupt in the top-middle and top-right panels. Each square in the grid represents a voxel which summates activity within its frame as shown in the bottom panels. For the smoother pattern of neural activity, the summation of each voxel (bottom left) captures the changing gradient from left to right depicted in the top-left, whereas for the less smooth representation in the middle panel all voxels sum to the same orange value (bottom middle). Thus, differences in activation of yellow vs. red neurons are detectable using fMRI for the smooth case, but not for the less smooth case because voxel response is homogenous. Improving spatial resolution (right panels) by reducing voxel size overcomes these sampling limits, resulting in voxel inhomogeneity (bottom-right panel).

rightmost column). Unfortunately, arbitrarily small voxels with high sampling rates is not a luxury afforded to fMRI.

The success of fMRI given its sampling limits is consistent with proposed neural coding schemes, such as population coding (*Averbeck et al., 2006*; *Panzeri et al., 2015*; *Pouget et al., 2000*) in cases where neurons with similar tunings spatially cluster (e.g., *Maunsell and Van Essen, 1983*). In population coding, neurons jointly contribute to represent a stimulus in much the same way as pixels were contributing to represent different colors in the leftmost column of *Figure 1*. When this inhomogeneity breaks down, similarity structures should be difficult to recover using fMRI. Indeed, a recent study with macaque monkeys which considered both single-cell and fMRI measures supports this viewpoint — stimulus aspects which were poorly spatially clustered in terms of single cell selectivity were harder to decode from the BOLD response (*Dubois et al., 2015*).

The same principles extend from the spatial to the temporal domain. The BOLD response will be blind to the differences between representations to the extent that the brain relies on the precise timing of neural activity to code information. For example, in burstiness coding, neural representations are distinguished from one another not by their average firing rate but by the variance of their activity (*Fano, 1947*; *Katz, 1996*). Under this coding scheme, more intense stimulus values are represented by burstier units, not units that fire more overall. Neural similarity is not recoverable by fMRI under a burstiness coding scheme. Because the BOLD signal roughly summates through time (*Boynton et al., 1996*), firing events will sum together to the same number irrespective of their burstiness.

Likewise, BOLD activity may be a composite of synchronized activity at multiple frequencies. Although gamma-band local field potential is most associated with BOLD response, oscillations at other frequency bands may also contribute to the BOLD response (*Magri et al., 2012*; *Scheeringa et al., 2011*). If so, fMRI would fail to distinguish between representational states that are differentiated by the balance of contributions across bands, much like the arithmetic example at the start of this subsection in which different addends yield the same sum. As before, basic concepts in information theory, such as the Nyquist-Shannon sampling theorem, imply that temporally demanding coding schemes will be invisible to fMRI (cf. *Nevado et al., 2004*).

The success of fMRI does not imply that the brain does not utilize precise timing information, but it does mean that such temporally demanding coding schemes cannot be the full story given fMRI's successes in revealing neural representations. Instead, the neural code must include in its mixture at least some coding schemes that are consistent with fMRI's successes. For example, rate coding (*Adrian and Zotterman, 1926*) in which the frequency at which neurons fire is a function of the intensity of a stimulus is consistent with the success of fMRI because changes in firing rate for a population of cells should be recoverable by fMRI as more blood flows to more active cells (*O'Herron et al., 2016*).

These examples make clear that the neural code must be somewhat spatially and temporally smooth with respect to neural activity (which is several orders of magnitude smaller than voxels) for fMRI to be successful. Whatever is happening in the roughly one million neurons within a voxel (*Huettel et al., 2009*) through time is being partially reflected by the BOLD summation, which would not be the case if each neuron was computing something dramatically different for in-depth discussion, see: (*Kriegeskorte et al., 2010*).

## Functional smoothness

One general conclusion is that important aspects of the neural code are spatially and temporally smooth. In a sense, this notion of smoothness is trivial as it merely implies that changes in neural activity must be visible in the BOLD response (i.e., across-voxel inhomogeneity) for fMRI to be successful. In this section, we focus on a more subtle sense of smoothness that must also be satisfied, namely functional smoothness.

Neighboring voxels predominantly contain similar representations (*Norman et al., 2006*), i.e., they are topologically organized like in *Figure 1*. However, *super-voxel smoothness* is neither necessary nor sufficient for fMRI to succeed in recovering similarity structure. Instead, a more general notion of functional smoothness must be satisfied in which similar stimuli map to similar internal representations. Although super-voxel and functional smoothness are both specified at the super-voxel level, these distinct concepts should not be confused. A function $f$ that maps from some input $x$ to output $y$ is functionally smooth if and only if

$$\text{sim}(x_1, x_2) \propto \text{sim}(y_1, y_2), \tag{1}$$

where $f(x_1) = y_1$ and $f(x_2) = y_2$. For example, $x$ and $y$ could be the beta estimates for voxels in two brain regions and sim could be Pearson correlation. To measure functional smoothness, the degree of proportionality between all possible similarity pairs $\text{sim}(x_1, x_2)$ and $\text{sim}(y_1, y_2)$ also could be assessed by Pearson correlation (i.e., does the similarity between a $y_1$ and $y_2$ pair increase as the similarity increases between the corresponding $x_1$ and $x_2$ pair). By definition, functional smoothness needs to be preserved in the neural code for fMRI to recover similarity correspondences (as in RSA), whether these correspondences are between stimuli (e.g., $x$s) and neural activity (e.g., $y$s), multiple brain regions (e.g., $x$s and $y$s), or model measure (e.g., $x$s) and some brain region (e.g., $y$s).

From this definition, it should be clear that functional smoothness is distinct from super-voxel smoothness. For example, a brain area that showed smooth activity patterns across voxels for each individual face stimulus but whose activity did not reflect the similarity structure of the stimuli would be super-voxel smooth, but not functionally smooth with respect to the stimulus set. Conversely, a later section of this contribution discusses how neural networks with random weights are functionally but not super-voxel smooth.

To help introduce the concept of functional smoothness, we consider two coding schemes used in engineering applications, factorial and hash coding, which are both inconsistent with the success of fMRI because they do not preserve functional smoothness. In the next section, we consider coding schemes, such as deep learning networks, that are functionally smooth to varying extents and are consistent with the success of fMRI.

## Factorial design coding

Factorial design is closely related to the notion of hierarchy. For example, hierarchical approaches to human object recognition (*Serre and Poggio, 2010*) propose that simple visual features (e.g., a horizontal or vertical line) are combined to form more complex features (e.g., a cross). From a factorial perspective, the simple features can be thought of as main effects and the complex features, which reflect the combination of simple features, as interactions.

In *Table 1*, a $2^3$ two-level full factorial design is shown with three factors A, B, C, three two-way interactions AB, AC, BC, and a three-way interaction ABC, as well as an intercept term. All columns in the design matrix are pairwise orthogonal.

Applying the concept of factorial design to modeling the neural code involves treating each row in *Table 1* as a representation. For example, each entry in a row could correspond to the activity level of a voxel. Interestingly, if any region in the brain solely had such a distribution of voxels, neural similarity would be impossible to recover by fMRI. The reason for this is that every representation (i.e., row in *Table 1*) is orthogonal to every other row, which means the neural similarity is the same for any pair of items. Thus, this coding scheme cannot uncover that low distortions are more similar to a category prototype than high distortions.

Rather than demonstrate by simulation, we can supply a simple proof to make this case using basic linear algebra. Dividing each item in the $n \times n$ design matrix (i.e., *Table 1*) by $\sqrt{n}$, makes each

**Table 1.** Design matrix for a $2^3$ full factorial design.

| I | A | B | C | AB | AC | BC | ABC |
|---|---|---|---|----|----|----|-----|
| 1 | −1 | −1 | −1 | 1 | 1 | 1 | −1 |
| 1 | 1 | −1 | −1 | −1 | −1 | 1 | 1 |
| 1 | −1 | 1 | −1 | −1 | 1 | −1 | 1 |
| 1 | 1 | 1 | −1 | 1 | −1 | −1 | −1 |
| 1 | −1 | −1 | 1 | 1 | −1 | −1 | 1 |
| 1 | 1 | −1 | 1 | −1 | 1 | −1 | −1 |
| 1 | −1 | 1 | 1 | −1 | −1 | 1 | −1 |
| 1 | 1 | 1 | 1 | 1 | 1 | 1 | 1 |

column orthonormal, i.e., each column will represent a unit vector and be orthogonal to the other columns. This condition means that the design matrix is orthogonal. For an orthogonal matrix, $Q$, like our design matrix, the following property holds: $Q \times Q^T = Q^T \times Q = I$; where $Q^T$ is the transpose of $Q$ (a matrix obtained by swapping columns and rows), and $I$ is the identity matrix. This property of orthogonal matrices implies that rows and columns in the factorial design matrix are interchangeable, and that both rows and columns are orthogonal.

The internal representations created using a factorial design matrix do not cluster in ways that meaningfully reflect the categorical structure of the inputs. Due to the fact that each representation is created such that it is orthogonal to every other, there can be no way for information, correlations within and between categories, to emerge. Two inputs varying in just one dimension (i.e., pixel) would have zero similarity; this is inherently not functionally smooth. In terms of *Equation 1* and *Table 1*, an $x$ would be a three dimensional vector consisting of the values of A, B, and C, whereas its $y$ would be the entire corresponding row from the table. Setting aside the degenerate case of self-similarity, there is no proportional relationship between similarity pairs because all $y$ pairs have zero similarity. If the neural code for a region was employing a technique similar to factorial design, neuroimaging studies would never uncover similarity structures by looking at the activity patterns of voxels in that region.

### Hash function coding

Hash functions assign arbitrary unique outputs to unique inputs, which is potentially useful for any memory system be it digital or biological. However, such a coding scheme is not functionally smooth by design. Hashing inputs allow for a memory, a data store known as a hash table, that is content-addressable (*Hanlon, 1966*; *Knott, 1975*) — also a property of certain types of artificial neural network (*Hopfield, 1982*; *Kohonen et al., 1987*). Using a cryptographic hash function means that the arbitrary location in memory of an input is a function of the input itself.

We employed (using the procedure below) the secure cryptographic hash algorithm 1 (SHA-1), an often-used hash function, and applied it to each value in the input vector (*National Institute of Standards and Technology, 2015*). Two very similar inputs (e.g., members of the same category) are extremely unlikely to produce similar SHA-1 hashes. Thus, they will be stored distally to each other, and no meaningful within-category correlation will arise (i.e., functional smoothness is violated). Indeed, in cryptography applications any such similarities could be exploited to make predictions about the input.

If the neural code in a brain area was underpinned by behavior akin to that of a hash function, imaging would be unable to detect correlations with the input. This is due to the fact that hash functions are engineered in such a way as to destroy any correlations, while nonetheless allowing for the storage of the input in hash tables.

Although hash tables do not seem well-matched to the demands of cognitive systems that generalize inputs, they would prove useful in higher-level mental functions such as source memory monitoring. Indeed, to foreshadow a result below, the advanced layers of very deep artificial neural networks approximate a cryptographic hash function, which consequently makes it difficult to recover the similarity structure in those layers.

## Model

In this section, we consider whether neural networks with random weights are consistent with the success of fMRI given its limitations in measuring neural activity. Simulations in the next section revisit these issues through the lens of a deep learning model trained to classify photographs of real-world categories, such as moped, tiger, guitar, robin, etc.

Each simulation is analogous to performing fMRI on the candidate neural code. These simple simulations answer whether in principle neural similarity can be recovered from fMRI data taken from certain neural coding schemes. Stimuli are presented to a model while its internal representations are measured by a simulated fMRI scanner.

The methods were as follows. The stimuli consist of 100-dimensional vectors that were distortions of an underlying prototype. As noise is added to the prototype and the distortion level increases, the neural similarity (measured using Pearson's correlation coefficient $\rho$) between the prototype and

its member should decrease. The question is whether we can recover this change in neural similarity in our simulated fMRI scanner.

First, each fully-connected network was initialized to random weights drawn from a Gaussian distribution ($\mu = 0, \sigma = 1$). Then, a prototype was created from 100 draws from a Gaussian distribution ($\mu = 0, \sigma = 1$). Nineteen distortions of the prototype were created by adding levels of Gaussian noise with increasing standard deviation ($\sigma = \sigma_{prev} + 0.05$) to the prototype. Finally, each item was re-normalized and mean centered, so that $\mu = 0$ and $\sigma = 1$ regardless of the level of distortion. This procedure was repeated for 100 random networks. In the simulations that follow, the models considered involved some portion of the randomly-initialized 8-layer network ($8 \times 100^2$ weights).

The coding schemes that follow are important components in artificial neural network models. The order of presentation is from the most basic components to more complex configurations of networks. To foreshadow the results shown in *Figure 2*, fMRI can recover the similarity structure for all of these models to varying degrees with the simpler models faring better than the more complex models.

## Vector space coding

The first in this line of models considered is vector space coding (i.e., $\mathbb{R}^n$), in which stimuli are represented as a vector of real-valued features. Representing concepts in multidimensional spaces has a long and successful history in psychology (*Shepard, 1987*). For example, in a large space, lions and tigers should be closer to each other than lions and robins because they are more similar. The kinds of operations that are naturally done in vector spaces (e.g., additions and multiplications) are particularly well suited to the BOLD response. For example, the haemodynamic response to individual stimuli roughly summates across a range of conditions (*Dale and Buckner, 1997*) and this linearity seems to extend to representational patterns (*Reddy et al., 2009*).

In this neural coding scheme, each item (e.g., a dog) is represented as the set of values in its input vector (i.e., a set of numbers with range $[-1, 1]$). This means that for a given stimulus, the representation this model produces is identical to the input. In this sense, vector space coding is functionally smooth in a trivial sense as the function is identity. As shown in *Figure 2*, neural similarity gradually falls off with added distortion (i.e., noise). Therefore, this very simple coding scheme creates representational spaces that would be successfully detected by fMRI.

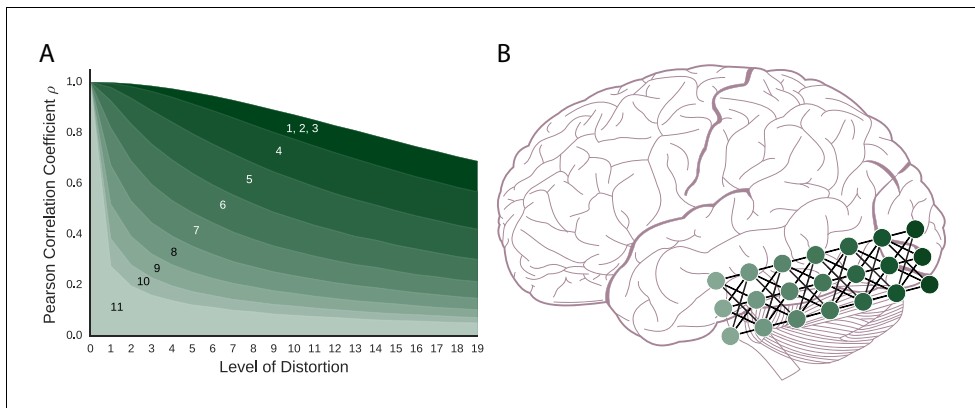

**Figure 2.** As models become more complex with added layers, similarity structure becomes harder to recover, which might parallel function along the ventral stream. (**A**) For the artificial neural network coding schemes, similarity to the prototype falls off with increasing distortion (i.e., noise). The models, numbered 1–11, are (*1*) vector space coding, (*2*) gain control coding, (*3*) matrix multiplication coding, (*4*), perceptron coding, (*5*) 2-layer network, (*6*) 3-layer network, (*7*) 4-layer network, (*8*) 5-layer network, (*9*) 6-layer network (*10*) 7-layer network, and (*11*), 8-layer network. The darker a model is, the simpler the model is and the more the model preserves similarity structure under fMRI. (**B**) A deep artificial neural network and the ventral stream can be seen as performing related computations. As in our simulation results, neural similarity should be more difficult to recover in the more advanced layers.

## Gain control coding

Building on the basic vector space model, this scheme encodes each input vector by passing it through a monotonic non-linear function, the hyperbolic tangent function ($\tanh$), which is functionally smooth. This results in each vector element being transformed, or squashed, to values between $[-1, 1]$. Such functions are required by artificial neural networks (and perhaps the brain) for gain control (*Priebe and Ferster, 2002*). The practical effect of this model is to push the values in the model's internal representation toward either $-1$ or $1$. As can be seen in *Figure 2*, neural similarity is well-captured by the gain control neural coding model.

## Matrix multiplication coding

This model performs more sophisticated computations on the input stimuli. In line with early connectionism and Rescorla-Wagner modeling of conditioning, this model receives an input vector and performs matrix multiplication on it, i.e., computes the weighted sums of the inputs to pass on to the output layer (*Knapp and Anderson, 1984*; *Rescorla and Wagner, 1972*). These simple one-layer neural networks can be surprisingly powerful and account for a range of complex behavioral findings (*Ramscar et al., 2013*). As we will see in later subsections, when a non-linearity is added (e.g., $\tanh$), one-layer networks can be stacked on one another to build deep networks.

This neural coding scheme takes an input stimulus (e.g., an image of a dog) and multiplies it by a weight matrix to create an internal representation, as shown in *Figure 3*. Interestingly, as shown in *Figure 3*, the internal representation of this coding scheme is completely nonsensical to the human eye and is not super-voxel smooth, yet it successfully preserves similarity structure (see *Figure 2*). Matrix multiplication maps similar inputs to similar internal representations. In other words, the result is not super-voxel smooth, but it is functionally smooth which we conjecture is critical for fMRI to succeed.

## Perceptron coding

The preceding coding scheme was a single-layer neural network. To create multi-layer networks, that are potentially more powerful than an equivalent single-layer network, a non-linearity (such as $\tanh$) must be added to each network layer post-synaptically. Here, we consider a single-layer network with the $\tanh$ non-linearity included (see *Figure 3*). As with matrix multiplication previously, this neural coding scheme is successful (see *Figure 2*) with 'similar inputs lead[ing] to similar outputs' (*Rummelhart et al., 1995*, p. 31).

## Multi-layered neural network coding

The basic network considered in the previous section can be combined with other networks, creating a potentially more powerful multi-layered network. These multi-layered models can be used to capture a stream of processing as is thought to occur for visual input to the ventral stream, shown in *Figure 2B* (*DiCarlo and Cox, 2007*; *Riesenhuber and Poggio, 1999*, *2000*; *Quiroga et al., 2005*; *Yamins and DiCarlo, 2016*).

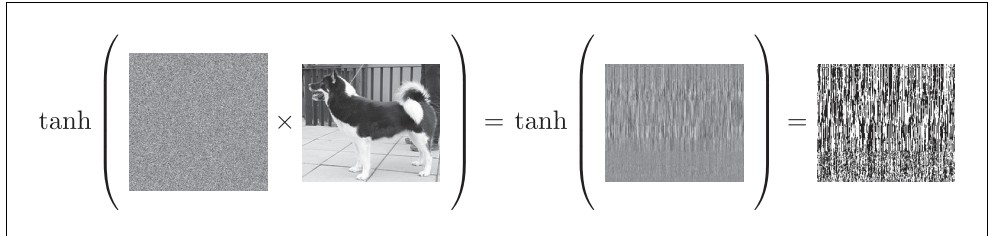

**Figure 3.** The effect of matrix multiplication followed by the $\tanh$ function on the input stimulus. The output of this one-layer network is shown, as well as the outcome of applying a non-linearity to the output of the matrix multiplication. In this example, functional smoothness is preserved whereas super-voxel smoothness is not. The result of applying this non-linearity can serve as the input to the next layer of a multi-layer network.

In this section, we evaluate whether the similarity preserving properties of single-layer networks extend to deeper, yet still untrained, networks. The simulations consider networks with 2 to 8 layers. The models operate in a fashion identical to the perceptron neural coding model considered in the previous section. The perceptrons are stacked such that the output of a layer serves as the input to the next layer. We only perform simulated fMRI on the final layer of each model. These simulations consider whether the representations that emerge in multi-layered networks are plausible given the success of fMRI in uncovering similarity spaces see also: (*Cox et al., 2015*; *Cowell et al., 2009*; *Edelman et al., 1998*; *Goldrick, 2008*; *Laakso and Cottrell, 2000*). Such representations, as found in deep artificial neural network architectures, are uncovered by adding layers to discover increasingly more abstract commonalities between inputs (*Graves et al., 2013*; *Hinton et al., 2006*; *Hinton, 2007*; *Hinton et al., 2015*; *LeCun et al., 2015*).

As shown in *Figure 2*, the deeper the network the less clear the similarity structure becomes. However, even the deepest network preserves some level of similarity structure. In effect, as layers are added, functional smoothness declines such that small perturbations to the initial input result in final-layer representations that tend to lie in arbitrary corners of the representational space, as the output takes on values that are $+1$ or $-1$ due to $\tanh$. As layers are added, the network becomes potentially more powerful, but less functionally smooth, which makes it less suitable for analysis by fMRI because the similarity space breaks down. In other words, two similar stimuli can engender near orthogonal (i.e., dissimilar) representations at the most advanced layers of these networks. We measured functional smoothness for a large set of random input vectors using *Equation 1* with Pearson correlation serving as both the similarity measure and measure of proportionality. Consistent with *Figure 2*'s results, at layer 1 (equivalent to the perceptron coding model) functional smoothness was $0.86$, but declined to $0.22$ by the eighth network layer. These values were calculated using all item pairs consisting of a prototype and one of its distortions. In the Discussion section, we consider the theoretical significance of these results in tandem with the deep learning network results (next section).

## Deep learning networks

Deep learning networks (DLNs) have led to a revolution in machine learning and artificial intelligence (*Krizhevsky et al., 2012*; *LeCun et al., 1998*; *Serre et al., 2007*; *Szegedy et al., 2015a*). DLNs outperform existing approaches on object recognition tasks by training complex multi-layer networks with millions of parameters (i.e., weights) on large databases of natural images. Recently, neuroscientists have become interested in how the computations and representations in these models relate to the ventral stream in monkeys and humans (*Cadieu et al., 2014*; *Dubois et al., 2015*; *Guclu and van Gerven, 2015*; *Hong et al., 2016*; *Khaligh-Razavi and Kriegeskorte, 2014*; *Yamins et al., 2014*; *Yamins and DiCarlo, 2016*). For these reasons, we choose to examine these models, which also allow for RSA at multiple representational levels.

In this contribution, one key question is whether functional smoothness breaks down at more advanced layers in DLNs as it did in the untrained random neural networks considered in the previous section. We address this question by presenting natural image stimuli (i.e., novel photographs) to a trained DLN, specifically Inception-v3 GoogLeNet (*Szegedy et al., 2015b*), and applying RSA to evaluate whether the similarity structure of items would be recoverable using fMRI.

### Architecture

The DLN we consider, Inception-v3 GoogLeNet, is a convolutional neural network (CNN), which is a type of DLN especially adept at classification and recognition of visual inputs. CNNs excel in computer vision, learning from huge amounts of data. For example, human-like accuracy on test sets has been achieved by: LeNet, a pioneering CNN that identifies handwritten digits (*LeCun et al., 1998*); HMAX, trained to detect objects, e.g., faces, in cluttered environments (*Serre et al., 2007*); and AlexNet, which classifies photographs into 1000 categories (*Krizhevsky et al., 2012*).

The high-level architecture of CNNs consists of many layers (*Szegedy et al., 2015a*). These are stacked on top of each other, in much the same way as the stacked multilevel perceptrons described previously. A key difference is that CNNs have more variety especially in breadth (number of units) between layers.

In many CNNs, some of the network's layers are convolutional, which contain components that do not receive input from the whole of the previous layer, but a small subset of it (*Szegedy et al., 2015b*). Many convolutional components are required to process the whole of the previous layer by creating an overlapping tiling of small patches. Often convolutional layers are interleaved with max-pooling layers (*Lecun et al., 1998*), which also contain tile-like components that act as local filters over the previous layer. This type of processing and architecture is both empirically driven by what works best, as well as inspired by the visual ventral stream, specifically receptive fields (*Fukushima, 1980*; *Hubel and Wiesel, 1959*, *1968*; *Serre et al., 2007*).

Convolutional and max-pooling layers provide a structure that is inherently hierarchical. Lower layers perform computations on small localized patches of the input, while deeper layers perform computations on increasingly larger, more global, areas of the stimuli. After such localized processing, it is typical to include layers that are fully-connected, i.e., are more classically connectionist. And finally, a layer with the required output structure, e.g., units that represent classes or a yes/no response as appropriate.

Inception-v3 GoogLeNet uses a specific arrangement of these aforementioned layers, connected both in series and in parallel (*Szegedy et al., 2015b*, *2015a*, *2016*). In total it has 26 layers and 25 million parameters inclusive of connection weights (*Szegedy et al., 2015b*). The final layer is a softmax layer that is trained to activate a single unit per class. These units correspond to labels that have been applied to sets of photographs by humans, e.g., 'space shuttle', 'ice cream', 'sock', within the ImageNet database (*Russakovsky et al., 2015*).

Inception-v3 GoogLeNet has been trained on millions of human-labeled photographs from 1000 of ImageNet's synsets (sets of photographs). The 1000-unit wide output produced by the network when presented with a photograph represents the probabilities of the input belonging to each of those classes. For example, if the network is given a photograph of a moped it may also activate the output unit that corresponds to bicycle with activation 0.03. This is interpreted as the network expressing the belief that there is a 3% probability that the appropriate label for the input is 'bicycle'. In addition, this interpretation is useful because it allows for multiple classes to co-exist within a single input. For example, a photo with a guillotine and a wig in it will cause it to activate both corresponding output units. Thus the network is held to have learned a distribution of appropriate labels that reflect the most salient items in a scene. Inception-v3 GoogLeNet, achieves human levels of accuracy on test sets, producing the correct label in its five most probable guesses approximately 95% of the time (*Szegedy et al., 2015b*).

## Deep learning network simulation

We consider whether functional smoothness declines as inputs are processed by the more advanced layers of Inception-v3 GoogLeNet. If so, fMRI should be less successful in brain regions that instantiate computations analogous to the more advanced layers of such networks. Unlike the previous simulations, we present novel photographs of natural categories to these networks. The key question is whether items from related categories (e.g., banjos and guitars) will be similar at various network layers. The 40 photographs (i.e., stimuli) are divided equally amongst 8 subordinate categories: banjos, guitars, mopeds, sportscars, lions, tigers, robins, and partridges, which in turn aggregate into 4 basic-level categories: musical instruments, vehicles, mammals, and birds; which in turn aggregate into 2 superordinates: animate and inanimate.

We consider how similar the internal network representations are for pairs of stimuli by comparing the resulting network activity, which is analogous to comparing neural activity over voxels in RSA. Correlations for all possible pairings of the 40 stimuli were calculated for both a mid and a later network layer (see *Figure 4*).

The middle layer (*Figure 4A*) reveals cross-category similarity at both the basic and superordinate level. For example, lions are more like robins than guitars. However, at the later layer (*Figure 4B*) the similarity structure has broken down such that subordinate category similarity dominates (i.e., a lion is like another lion, but not so much like a tiger). Interestingly, the decline in functional smoothness is not a consequence of sparseness at the later layer as the Gini coefficient, a measure of sparseness (*Gini, 1909*), is 0.947 for the earlier middle layer (*Figure 4A*) and 0.579 for the later advanced layer (*Figure 4B*), indicating that network representations are distributed in general and even more so at the later layer. Thus, the decline in functional smoothness at later layers does not appear to be a straightforward consequence of training these networks to classify stimuli, although

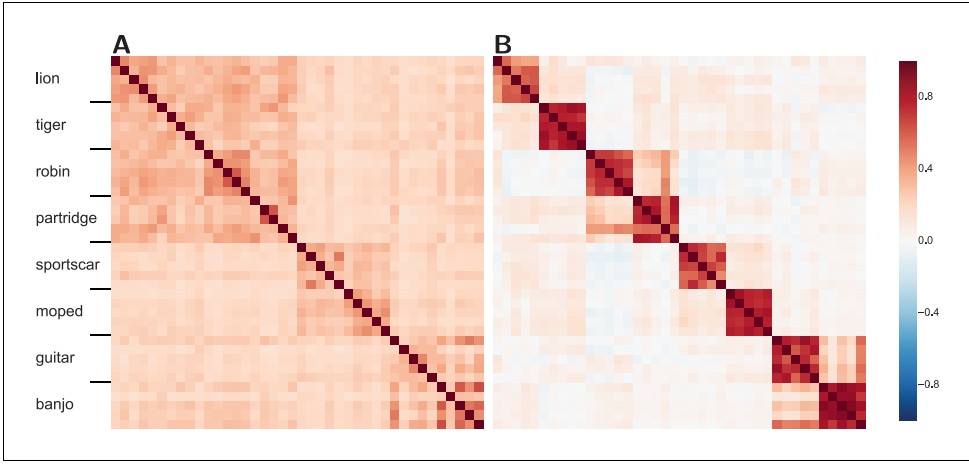

**Figure 4.** Similarity structure becomes more difficult to recover in the more advanced layers of the DLN. (**A**) The similarity structure in a middle layer of a DLN, Inception-v3 GoogLeNet. The mammals (lions and tigers) and birds (robins and partridges) correlate forming a high-level domain, rendering the upper-left quadrant a darker shade of red. Whereas the vehicles (sportscars and mopeds) and musical instruments (guitars and banjos) form two high-level categories. (**B**) In contrast, at a later layer in this network, the similarity space shows high within-category correlations and weakened correlations between categories. While some structure between categories is preserved, mopeds are no more similar to sportscars than they are to robins.

it would be interesting to compare to unsupervised approaches that can perform at equivalent accuracy levels (no such network currently exists).

These DLN results are directly analogous to those with random untrained networks (see *Figure 2*). In those simulations, similar input patterns mapped to orthogonal (i.e., dissimilar) internal representations in later layers. Likewise, the trained DLN at later layers can only capture similarity structure within subordinate categories (e.g., a tiger is like another tiger) which the network was trained to classify. The effect of training the network was to create equivalence classes based on the training label (e.g., tiger) such that members of that category are mapped to similar network states. Violating functional smoothness, all other similarity structure is discarded such that a tiger is no more similar to a lion than to a banjo from the network's perspective. Should brain regions operate in a similar fashion, fMRI would not be successful in recovering similarity structure therein. In the Discussion, we consider the implications of these findings on our understanding of the ventral stream and the prospects for fMRI.

## Discussion

Neuroscientists would rightly prefer a method that had both excellent spatial and temporal resolution for measuring brain activity. However, as we demonstrate in this article, the fact that fMRI has proven useful in examining neural representations, despite limitations in both its temporal and spatial resolution, says something about the nature of the neural code. One general conclusion is that the neural code must be smooth, both at voxel (such that voxel responses are inhomogeneous across time and space) and functional levels.

The latter notion of smoothness is often overlooked or confused with super-voxel smoothness, but is necessary for fMRI to recover similarity spaces in the brain. Coding schemes, such as factorial and hash coding, are useful in numerous real-world applications and have an inverse function (i.e., one can go backwards from the internal representation to recover the unique stimulus input). However, these schemes are incompatible with the success of fMRI because they are not functionally smooth. For example, if the brain solely used such coding schemes, the neural representation of a robin would be no more similar to that of a sparrow than to that of a car. The fact that such neural similarities are recoverable by fMRI suggests that the neural code differs from these schemes in many cases.

In contrast, we found that the types of representations used and generated by artificial neural networks, including deep learning networks, are broadly compatible with the success of fMRI in assessing neural representations. These coding schemes are functionally smooth in that similar inputs tend toward similar outputs, which allows item similarity to be reflected in neural similarity (as measured by fMRI). However, we found that functional smoothness breaks down as additional network layers are added. Specifically, we have shown that multi-layer networks eventually converge to something akin to a hash function, as arbitrary locations in memory correspond to categories of inputs.

These results take on additional significance given the recent interest in deep artificial networks as computational accounts of the ventral stream. One emerging view is that the more advanced the layers of these models correspond to more advanced regions along the ventral stream (*Cadieu et al., 2014-12*; *Dubois et al., 2015*; *Guclu and van Gerven, 2015*; *Hong et al., 2016*; *Khaligh-Razavi and Kriegeskorte, 2014*; *Yamins et al., 2014*; *Yamins and DiCarlo, 2016*).

If this viewpoint is correct, our results indicate that neural representations should progressively become less functionally smooth and more abstract as one moves along the ventral stream (recall *Figure 2*). Indeed, neural representations appear to become more abstract, encoding whole concepts or categories, as a function of how far along the ventral stream they are located (*Bracci and Op de Beeck, 2016*; *DiCarlo and Cox, 2007*; *Riesenhuber and Poggio, 1999*, *2000*; *Yamins and DiCarlo, 2016*). For example, early on in visual processing, the brain may extract so-called basic features, such as in broadly-tuned orientation columns (*Hubel and Wiesel, 1959*, *1968*). In contrast, later on in processing, cells may selectively respond to particular individual stimulus classes i.e., Jennifer Aniston, grandmother, concept, or gnostic cells (*Gross, 2002*; *Konorski, 1967*; *Quiroga et al., 2005*), irrespective of orientation, etc.

Likewise, we found that Inception-v3 GoogLeNet's representations became symbol-like at advanced network layers such that items sharing a category label (e.g., tigers) engendered related network states, while items in other categories engendered orthogonal states (recall *Figure 4*). Our simulations of random networks also found reduced functional smoothness at advanced network layers, suggesting a basic geometric property of multi-layer networks. The effect of training seems limited to creating network states in which stimuli that share the same label (e.g., multiple viewpoints of Jennifer Aniston) become similar and items from all other categories (even if conceptually related) become orthogonal. If so, areas further along the ventral stream should prove less amenable to imaging (recall *Figure 2*). Indeed, a recent object recognition study found that the ceiling on observable correlation values becomes lower as one moves along the ventral stream (*Bracci and Op de Beeck, 2016*).

Here, we focused on using fMRI to recover non-degenerate similarity spaces (i.e., where there are similarities beyond self-similarities). However, functional smoothness is also important for other analysis approaches. For example, MVPA decoders trained to classify items (e.g., is a house or a face being shown?) based on fMRI activity will only generalize to novel stimuli when functional smoothness holds. Likewise, univariate clusters (e.g., a house or face area) will most likely be found and generalize to novel stimuli when functional smoothness holds because functional smoothness implies similar activation profiles for similar stimuli. Functional smoothness should be an important factor in determining how well classifiers perform and how statistically robust univariate clusters of voxels are.

In cognitive science, research is often divided into levels of analysis. In Marr's levels, the top level is the problem description, the middle level captures how the problem is solved, and bottom level concerns how the solution is implemented in the brain (*Marr, 1982*). Given that the 'how' and 'where' of cognition appear to be merging, some have questioned the utility of this tripartite division (*Love, 2015*).

Our results suggest another inadequacy of these three levels of description, namely that the implementation level itself should be further subdivided. What is measured by fMRI is at a vastly more abstract scale than what can be measured in the brain. For example, major efforts, like the European Human Brain Project and the Machine Intelligence from Cortical Networks project (*Underwood, 2016*), are chiefly concerned with fine-grained aspects of the brain that are outside the reach of fMRI (*Chi, 2016*; *Frégnac and Laurent, 2014*). Likewise, models of spiking neurons e.g., (*Wong and Wang, 2006*) are at a level of analysis lower than where fMRI applies.

Nevertheless, fMRI has proven useful in understanding neural representations that are consequential to behavior. Perhaps this success suggests that the appropriate level for relating brain to behavior is close to what fMRI measures. This does not mean lower-level efforts do not have utility

when the details are of interest. However, fMRI's success might mean that when one is interested in the nature of computations carried out by the brain, the level of analysis where fMRI applies may be preferred. To draw an analogy, one could construct a theory of macroeconomics based on quantum physics, but it would be incredibly cumbersome and no more predictive nor explanatory than a theory that contained abstract concepts such as money and supply. Reductionism, while seductive, is not always the best path forward.

## Acknowledgement

This work was supported by the Leverhulme Trust (Grant RPG-2014-075), the NIH (Grant 1P01HD080679), and a Wellcome Trust Investigator Award (Grant WT106931MA) to BCL, as well as The Alan Turing Institute under the EPSRC grant EP/N510129/1. Correspondences regarding this work can be sent to o.guest@ucl.ac.uk or b.love@ucl.ac.uk. The authors declare that they have no competing interests. The authors would like to thank Sabine Kastner, Russ Poldrack, Tal Yarkoni, Niko Kriegeskorte, Sam Schwarzkopf, Christiane Ahlheim, and Johan Carlin for their thoughtful feedback on the preprint version, 10.1101/071076. The code used to run these experiments is freely available, http://osf.io/v8baz.

## Additional information

### Funding

| Funder | Grant reference number | Author |
| --- | --- | --- |
| Leverhulme Trust | RPG-2014-075 | Bradley C Love |
| Wellcome | WT106931MA | Bradley C Love |
| National Institutes of Health | 1P01HD080679 | Bradley C Love |

The funders had no role in study design, data collection and interpretation, or the decision to submit the work for publication.

### Author contributions

OG, Conceptualization, Data curation, Software, Formal analysis, Visualization, Writing—original draft, Writing—review and editing; BCL, Conceptualization, Resources, Supervision, Funding acquisition, Writing—original draft, Project administration, Writing—review and editing

### Author ORCIDs

Olivia Guest, http://orcid.org/0000-0002-1891-0972
Bradley C Love, http://orcid.org/0000-0002-7883-7076

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
