## [Decision Letter]

Thank you for submitting your article "What the Success of Brain Imaging Implies about the Neural Code" for consideration by *eLife*. Your article has been reviewed by three peer reviewers, and the evaluation has been overseen by a Reviewing Editor and Sabine Kastner as the Senior Editor. The following individuals involved in review of your submission have agreed to reveal their identity: Rogier Andrew Kievit (Reviewer #1); Stefano Panzeri (Reviewer #2).

The reviewers have discussed the reviews with one another and the Reviewing Editor has drafted this decision to help you prepare a revised submission.

Summary:

This paper asks how the success of fMRI (particularly in the context of representational analysis) has implications for the structure of the neural code. The authors first argue that neural representations must be smooth enough in the spatial and temporal domains to afford effective measurement with fMRI. They then argue that the neural code must also obey a principle of "functional smoothness", such that similar stimuli map to similar neural representations. Using a range of coding schemes, they examine which ones could potentially satisfy this requirement for functional smoothness and thus allow successful neural decoding from fMRI. They show that a number of simple neural models demonstrate such representational isomorphism; in particular, they demonstrate at the earlier layers of a deep convolutional neural network (CNN) show functional similarity, whereas the later layers in the hierarchy do not. The authors conclude that the success of fMRI places limits on the range of plausible coding schemes, and in particular that it implies that neural coding schemes must be both spatiotemporally and functionally smooth.

Essential revisions:

The three reviewers and I all agree that this paper raises an interesting set of questions. However, there were a number of concerns about the paper as it is currently written. Here are the main concerns as I see them, and how they should be addressed:

– Two reviewers raise the question of whether your claims are fully supported by the results, suggesting in many cases that you are making strong claims that are not fully supported by your results. I think that in general the revision needs to better calibrate the claims to the evidence, and avoid overreaching. In particular, the fact that fMRI is successful does not mean that it is capturing all, or even the majority, of interesting neural signals. This issue needs to be clearly addressed in the revision.

– The concept of "functional smoothness" is central to your manuscript, but the reviewers raised multiple questions regarding the clarity of this concept in your paper. The revision needs to be much clearer about what this concept means and how it relates more specifically to the other concepts of smoothness.

– Reviewer 2 points out that your paper needs to make better contact with the literature that has investigated the neural basis of the BOLD signal. In particular, specific attention must be paid to the points raised by reviewer 2 regarding the implications for temporal aspects of the BOLD signal.

– Reviewer 3 points out that the paper lacks methodological details, and that they were not able to access the materials linked at OSF. In the revision, please include a detailed description of your methods along with a working link to the code and materials.

I am including the reviewers' comments below, which provide greater detail regarding the issues outlined above.

Reviewer #1:

In this paper, Guest and Love propose an interesting and provocative claim: The success of fMRI writ large can, and should, constrain theories about the underlying neural code. The paper is engaging and provides a considerable amount of food for thought. However, upon close reading a number of problems are present, including the scope and precision of the central claims, the use and definition of terminology, the choice of models and appropriate engagement with relevant literature. These problems are outlined below. Given the scope of the paper the comments are lengthy, but necessary to clearly describe what we found challenging about the manuscript.

1) What are the scopes of the claims being made?

One main issue with the paper is the precise nature of the central claim. This claim varies in scope throughout the paper from more modest and well-evidenced to more all-encompassing and arguably beyond the evidence put forth. The main claim can loosely be phrased as: The fact that fMRI writ large “works” tells us something about the underlying neural mechanisms. The precise cashing out of this claim includes the following descriptions:

Abstract “certain neural coding schemes are more likely than others” (Plausible but not necessarily true).

Subsection “Sub-voxel Smoothness”, “the rate of change (i.e., smoothness) of neural activity must not exceed what can be measured within a voxel.” (This phrasing seems overly general – There is no reason why rates of change couldn't also exceed what can be measured. All that is compellingly shown is that a non-trivial part of the neural code behaves in a temporally smooth manner).

In the same section “temporally demanding coding schemes cannot carry the day” (Not clear what “carry the day” means – If it means “Can't fully explain all patterns” then it is likely true and well supported, if it means “temporally demanding coding schemes cannot play an important role also” I think it's too strong a claim).

Also in this section “the neural code must be spatially and temporally smooth with respect to neural activity” (Overly general in that it implies there is only one neural code ('The')).

Discussion section first paragraph “the neural code must be smooth, both at the sub-voxel and functional levels.” (overly general – It can be smooth and non-smooth simultaneously to currently unknown degrees, e.g. Figure 1 left and right could be overlapping).

Discussion section paragraph ten “However, fMRI's success might mean that when one is interested in the nature of computations carried out by the brain, the level of analysis where fMRI applies should be preferred.” (A considerable overgeneralisation – All we know for sure is that fMRI can provide important information, but there is no reason why lower (or higher) levels of temporal or spatial abstraction can't be even more informative.).

More limited and arguably more accurate are:

– Abstract section “Deep neural network approaches, are consistent with the success of fMRI” (True and well-supported).

– Subsection “Sub-voxel Smoothness”, “The success of fMRI does not imply that the brain does not utilize precise timing information” (an important explication of the point raised above that multiple coding schemes could operate concurrently, and that fMRI would 'work' if it only picks up parts).

Similarly in subsection “Factorial Design Coding”, discussing factorial coding, the authors state “Interestingly, if any region in the brain had such a distribution of voxels, neural similarity would be impossible to recover by fMRI.” And later “If the neural code for a region was employing a technique similar to factorial design, neuroimaging studies would never recover similarity structures by looking at the patterns of active voxels in that region.” These statements are only true if other coding schemes aren't allowed to co-exist. It may well be that factorial coding schemes co-exist alongside spatially smooth codes but simply cannot be detected using current methods and/or analyses. Or, to couch it in terms of Figure 1, there is no reason why the neural code might not be an overlay, or mixture, of A and B. This implicit suggestion that there must be a single coding scheme is reflected in the title, which refers to 'The' neural code.

As can be seen when read in succession, these are not interchangeable epistemic claims – Some are 'merely' existence claims whereas others rule out alternative explanations.

In our view, the main claim of that article, that is important and well supported, can be phrased as follows:

The success of fMRI makes it highly likely that at least some non-trivial subset of the signal/computations made by the neural code must be smooth.

Or, in the authors words “One general conclusion is that important aspects of the neural code are spatially and temporally smooth at the sub-voxel level”.

In other words, the article provides solid evidence for a more modest positive claim but at times shifts into language that suggests it has ruled out (or rendered exceedingly unlikely) other neural codes. We do not believe this is the case (nor, for that matter, do I think this is an achievable goal in the context of any single paper, for both principled mathematical reasons that every data pattern is compatible with an infinity of data generating mechanisms, and the pragmatic reasons of scope). In other words, we would suggest that the precise scope of the central claim is delineated more clearly, and that this scope is retained throughout

2) Is "sub-voxel smoothness" an accurate and clear description of what is required for successful fMRI?

The other main challenge in the article is the introduction of two new terms, spatial and functional smoothness. It seems as though these are not entirely precisely defined, and to the extent that they are, one might wonder whether a) 'smoothness' is the best term and b) what, if anything, is similar about spatial and functional smoothness such that they share a conceptual term. We outline these challenges below.

With regards to spatial smoothness, I wonder whether the term inhomogeneity isn't more appropriate or central to the claim being made. For instance, it seems as though one could rearrange the voxel columns in Figure 1 whilst preserving sub-voxel smoothness as well as functional smoothness – The arrangement of the voxels as a gradient could be misunderstood as being about super-voxel smoothness (especially in the light of evidence for early visual retinotopic mapping). (more detail below).

The paper distinguishes between the concepts of sub-voxel and super-voxel smoothness. These concepts appear to be independent – i.e. one could have subvoxel smoothness combined with non-smooth super-voxel patterns. i.e., it's not entirely clear why “In such a spatially smooth representation, the transitions from red to yellow occur in progressive increments” follows necessarily from subvoxel smoothness? The later definition that voxel size can be changed arbitrarily does capture this, but for a *given* voxel size a pattern could be subvoxel smooth and super-voxel non-smooth. Note that the empirical examples outlines in subsection “Sub-voxel Smoothness” are compelling evidence that sub and super-voxel smoothness are related, but in principle they needn't be (this point is made by the authors in subsection “Functional Smoothness” with respect to functional smoothness).

It might be helpful to show in Figure 1 examples of all four possible combinations. In the current Figure 1, the left panels appear to show both sub- and super-voxel smoothness (there is a smooth gradient of functional tunings both within a voxel, and across voxels once tunings have been averaged within each voxel). The right panel appears to show a case where neither hold (within a single voxel, tunings are arranged into jagged repeating stripes, and across voxels all tunings are identical once averaged). It would also be possible to have sub-voxel smoothness without super-voxel smoothness (e.g. if one were to shuffle the "columns" of the arrangement on the left), and vice versa (e.g. a version of the arrangement on the right in which the red stripes became progressively thinner and the yellow stripes progressively wider as one moved across the cortical sheet).

Picturing these four alternatives, it seems that in order for different stimuli to create different patterns at the level of fMRI voxels, the important factor is neither sub-voxel nor super-voxel smoothness, but something which might be termed "across-voxel inhomogeneity" – i.e. that different voxels sample populations of neurons with different functional properties. For example, in the example just suggested, wherein the 'columns' of the left-hand panel are shuffled, there would no longer be a smooth gradient in the functional selectivities of voxels as one moves across the cortical sheet, but different stimuli could still evoke unique multi-voxel patterns.

It is important here to engage with previous literature on fMRI decoding, especially Kamitani & Tong (2005) and subsequent reflections on why stimulus orientation can be decoded from V1 via fMRI (e.g. Alink et al. 2013; Carlson, 2014). Orientation columns in V1 are arranged in a similar fashion to the "sub-voxel non-smooth" depiction in Figure 1, yet orientation can reliably be decoded. One plausible reason is that the chance of each voxel sampling all orientation selectivities exactly equally is very low, so even if neuronal selectivities are intermixed at a sub-voxel level, slight differences in orientation preference are likely to emerge at the voxel level.

The caption of Figure 1 suggests non-smoothness is preserved “regardless of the precise boundaries of the voxels which quantize the brain” but it seems that this would only be true conditional on preserving approximate voxel size? If I am allowed to change both the size and boundaries I could create mostly-yellow and mostly-red voxels from Figure 1? Moreover, as mentioned above, random variations in sampling can lead to successful decoding of schemes similar to the non-smooth pattern.

Sub-voxel smoothness might not be the most intuitive term to use, not only for the reasons mentioned under point 2, but also because it implies spatial and not temporal smoothness. It seems more appropriate to talk about spatial and temporal scales at which useful information (about stimuli, thoughts, actions) is represented by brain activity.

3) "Functional smoothness" is a relational property between two models or between data and a model, not an inherent property of one model.

Arguably the most challenging term introduced by the paper is "functional smoothness." It is not made clear what this is a property of. From the definition, that "similar stimuli map to similar representations," a model is "functionally smooth" with respect to some second representational model if it preserves the representational geometry of that target model. In the first set of simulations involving noise-corrupted images, the target model is pixel space. In the second set of simulations involving objects of different categories being presented to a deep neural network, the target model is a semantic or conceptual space. In other words, functional smoothness is not an objective property of the model, but a relational property between an input and an output space. This seems to be recognised in some sections, e.g. subsection “Vector Space Coding”: "vector space coding is functionally smooth in the trivial sense as the function is identity."

However, many other phrases in the paper imply that smoothness is intrinsic to models, e.g.:

– subsection “Deep Learning Networks”: "one key question is whether functional smoothness breaks down at more advanced layers in DLNs…"

– subsection “Deep Learning Network Simulation”: "We consider whether functional smoothness declines as stimuli progress to more advanced layers…"

– and: "Violating functional smoothness, all other similarity structure is discarded…"

The simulations underlying Figure 2 and Figure 3 add to this confusion rather than illustrating functional smoothness. Presumably the weights within the weight matrices used in the models from "matrix multiplication coding" onwards were random? If so, then Figure 3 seems to illustrate the trivial effect that as one performs increasingly many random nonlinear transformations on an input, distances between stimuli in the input space will be less and less well correlated with distances in the output. This result doesn't seem to reveal anything fundamental about the merit of (non-random) deep nonlinear networks as models of brain representation.

To highlight the fact that functional smoothness is a relational property, it might instead be helpful to create a simulation akin to these, but using a trained network, and adding noise in a more abstract "target space" (e.g. noise could be added to the location of an object within an image, for example by sliding around the location of a dog image superimposed on a natural background). The earliest layer of the network should be "functionally smooth" w.r.t. changes in pixels but not location (i.e. pixelwise differences will make a large difference to early representations); middle layers may be "functionally smooth" w.r.t. location but not pixels (e.g. the same dog at nearby locations will activate the same feature detectors looking for eyes, legs, etc), while the final categorical layer would ideally be wholly invariant both to pixel and location changes provided the image continues to contain a dog.

It would also help to distinguish between coding schemes that have the capacity to be functionally smooth, vs. those that actually functionally smooth, with respect to a particular input space. Factorial and hash coding are raised as examples of codes that are "not functionally smooth," while neural-network-style encodings of various complexity are evaluated for whether the simulated examples happen to be functionally smooth with respect to the input spaces (pixel space, then semantic similarity space). However, there is an important difference in the sense in which factorial coding is "not functionally smooth" and that in which the late layers of GoogLeNet are not – factorial and hash coding are not capable of being functionally smooth (because every representation is orthogonal to every other), whereas the neural networks considered are capable in principle of strongly preserving the representational geometry of any input space desired (although see Point 6).

4) What is the relationship, if any, between spatial and functional smoothness?

It seems as if spatial and functional smoothness, as defined in the paper, should be completely independent of one another, and they are described in paragraph two of subsection “Functional Smoothness” as "distinct concepts." However:

Subsection “Matrix Multiplication Coding”: "…the internal representation of this coding scheme is completely nonsensical to the human eye and is not super-voxel smooth."

This is confusing for two reasons. First, there is no reason for any internal representation to be "sensible to the human eye." The fact that the input representation is "sensible" is just an artefact of using images as the inputs. Even then, the image is only understandable by the eye if one preserves the exact spatial order of units and arranges them into rows and columns of exactly the right dimensions. Any shuffling or re-arrangement of the pixels would constitute an identical representation, and yet would no longer be sensible to the eye – so sensibility to the eye does not seem relevant.

Second, how does this minimal model (a transformation of the input by a single matrix multiplication) specify properties about the spatial arrangement of voxels? Paragraph two subsection “Functional Smoothness” states that functional smoothness is defined at the level of voxels (although this seems a little counter-intuitive, since brains and neural networks encode information at the level of neurons). So in the "matrix multiplication code," we should imagine a case where there are as many voxels in the output as there are pixels in the input image, and the activation level of each one is determined by an arbitrary linear combination of all input pixels. This output will be "functionally smooth" w.r.t. pixel space if the matrix transformation is one that preserves representational geometry (e.g. a rotation matrix; see first comment under "Smaller points" below for more on this…). This will be true however one arranges the voxels spatially. Some possible arrangements will appear super-voxel smooth (e.g. if voxels are placed next to those with the most similar selectivities), and some will not (e.g. if voxels are randomly placed), but all arrangements will be functionally smooth.

If there is some deeper connection between functional and spatial smoothness, this needs to be more clearly explained and illustrated.

5) Different neural code properties may be required for "successful fMRI" when doing mean-activation vs. decoding vs. representational similarity analyses.

The title-bearing central claim refers to the “success” of brain imaging, but it is not entirely clear how this should be conceived. It might be worth briefly describing what counts as success, and how each result constrains possible neural codes. For example, it would be good to separately consider what findings from (1) old school mean-activation “blobology”, (2) classifiers performing above chance, (3) RSA imply about neural coding. It seems uncontroversial that all require "temporal smoothness" and some form of across-voxel spatial inhomogeneity in order for different stimuli to create detectably different fMRI activations. But they may have different further implications, for example:

1) The success of "blobology" in finding multi-voxel clusters with similar functional properties suggests some degree of super-voxel smoothness.

2) Above-chance decoding does not seem to even require a neural code capable of functional smoothness. E.g. in a factorial code, although every stimulus elicits a pattern orthogonal to that elicited by any other, one could still do successful "mind-reading" as long as one had access to data from previous trials on which subjects had viewed that stimulus.

3) The success of RSA (i.e. finding interesting and nuanced similarity patterns between patterns evoked by different stimuli, which seem to bear some relation to the geometry of those stimuli within other models such as pixel space or semantic space and can be compared to predictions from computational models) does require a neural code which is capable of functional smoothness. The importance of functional smoothness only to RSA does seem to be recognised in the paper, but could be made more explicit, e.g.:

Subsection “Factorial Design Coding” paragraph five: "If the neural code for a region was employing a technique similar to factorial design, neuroimaging studies would never recover similarity structures by looking at the patterns of active voxels in that region."

One thing to note here is that model comparisons do not necessarily require RSA. Does the success of other analysis methods that compare models (e.g. voxel-receptive-field modelling) also point to functional smoothness? Or is this term strongly linked to RSA as an analysis framework, to the extent that it does not have a meaning outside it? If so, why do the authors focus so strongly on functional smoothness? (RSA is successful, but so are (linear) classifiers).

6) Simulations with a network trained on a task other than categorisation would help justify the claim that "non-smoothness" is an inevitable property of deep nonlinear neural networks.

The final simulations, in which distances within a “semantic space” are informally compared to distances within successive layers of the deep neural network GoogLeNet, conclude that later layers are less functionally smooth w.r.t. semantic space than early ones (since they lose between-category similarity information), and that “the decline in functional smoothness at later layers does not appear to be a straightforward consequence of training these networks to classify stimuli.” This latter conclusion is likely to be controversial, and is not strongly supported by the sparsity analysis.

The simplest way to clarify the contribution of the training task to the representational geometry in the final layers would be to show RDMs from (a) randomly-weighted networks, and (b) an unsupervised network, or one trained on a task orthogonal to categorisation. A good candidate would be the unsupervised seven layer neural network in Wang & Gupta (2015), which is available from https://github.com/xiaolonw/caffe-video_triplet

Again though, this would not reveal anything about the "functional smoothness of the model," since that is not an inherent property, but only the similarity between the representations within the model and in semantic space.

7) Previous work

One key feature missing from this paper is closer engagement with previous literature on decoding, such as the seminal findings in Kamitani & Tong (2005) (discussed above in Point 2), computational accounts (e.g. Kriegeskorte, N., Cusack, R., & Bandettini, P. (2010) or de Beeck, H. P. O. (2010), and those discussing the plausibility of more trivial structural explanations such as differences in vasculature (e.g. Shmuel, A., Chaimow, D., Raddatz, G., Ugurbil, K., & Yacoub, E. (2010)).

Miscellaneous

Subsection “Matrix Multiplication Coding” states “Matrix multiplication maps similar inputs to similar internal representations.” The claim seems to refer to multiplication of a 1xn input by an arbitrary nxn matrix (such as would be implemented by a 1-layer fully connected linear neural network). Although there are some such matrix multiplications which would perfectly preserve distances between different inputs in their original vs. transformed spaces (e.g. rotation matrices), and with random matrices, distances in the original and transformed spaces will tend to correlate (as your simulations show), the claim is not generally true. There are many matrix multiplications which will completely disrupt representational geometry.

Subsection “Deep Learning Network Simulation” paragraph three states that sparseness of representation does not decline for a "later advanced layer" of the category-supervised deep neural net. Which layer is this? It seems surprising that sparseness does not increase in at least the final layer (i.e. the output of the softmax operation). Relatedly, is it worth showing more layers in Figure 5? If not, why are these two layers shown?

Related to Point 3 above, it would help to use more precise language about the nature of the sensory inputs or brain representations being discussed. For example, subsection “Matrix Multiplication Coding” says that a particular coding scheme "takes an input stimulus (e.g. a dog) and multiplies it by a weight matrix" – given that a dog is not the sort of thing that can be multiplied by a weight matrix, does this mean either (specifically) an image of a dog, or (generally) the activity within a preceding layer of neurons in a neural network model? Referring to the (arbitrary) input images in the simulations as "prototypes" is also confusing, as it suggests they have some special status to the models.

– Although I like the hash coding example in subsection “Hash Function Coding” says it's “potentially useful for biological systems” – it might be worth elaborating briefly in what circumstances a biological system would evolve something akin to hashcoding for certain stimuli? It seems rather inefficient and hard to reconcile with basic facts about learning (e.g. co-occurrence increasing association strengths, and thereby similarity) and memory (e.g. semantic co-activation)? I appreciate the later evidence for the compatibility of higher layers with hash coding but the above claim seems more general – This relates to the above discussion on what

– The selection of coding schemes cover interesting (and rhetorically convincing) ground but the motivation for this set of coding schemes doesn't seem to be motivated. E.g., why focus on hash coding and factorial mapping – Are those a subset of a broader range that could be considered? Something similar could be said about the selection of NN algorithms. The selection of codes seems to constitute an input being transformed by successively more complex neural networks ("vector space coding" = no transformation of the input; "gain control coding" = one nonlinearity; "matrix multiplication coding" = one linear transformation; "perceptron coding" = one linear transformation plus one nonlinearity; "multi-layer neural network"….etc). Although this is logical, the descriptions (e.g. "vector space coding") misleadingly imply that these are qualitatively distinct strategies for encoding a stimulus, and leave it to the reader to discover the logic of selecting these particular "codes".

– The paper by Bracci & de Boeck (2016) seems worth discussing in more detail, as it provides potential direct evidence for the hierarchy of smoothness. One wonders whether there are plausible alternative explanations that should be taken into account wrt varying levels of prediction accuracy across the ventral stream? For instance, the noise ceiling also often goes down (i.e. there is less signal to be explained in principle).

Reviewer #2:

This paper addresses an interesting subject, that of what we can learn about the neural code from fMRI. The paper makes a valuable conceptual effort to think about which neural codes are supported by fMRI observations and which are not. Much as I like the paper and I think it is important to discuss these issues, I think that the connection between neural activity and fMRI, which should be central to this topic, is not sufficiently well discussed. My fear is that places of the current manuscript would look insufficiently developed to neurophysiologists investigating neural coding. In the following I raise the attention of the authors to what are in my view problems in the current manuscript that need addressing, and I provide a few suggestions. I hope that this will improve their paper.

The current writing of the paper may be taken at specific places to argue that the success of fMRI implies that a coding scheme that does not come though fMRI is not one used by the brain to compute. ("Through proof and simulation, we determine which coding schemes are plausible given both fMRI's successes and its limitations in measuring neural activity"… "The neural code must have certain properties for this state of affairs to hold. What kinds of models or computations are consistent with the success of fMRI?" … "The success of fMRI does not imply that the brain does not utilize precise timing information, but it does mean that such temporally demanding coding schemes cannot carry the day given the successes fMRI has enjoyed in understanding neural representations.")

Of course there would be no basis for such a strong claim, and the authors should state and discuss this clearly. It is for example possible that fMRI gets only a part of the neural code used by the brain, and that other parts, perhaps as important as others, are simply lost by the limitations of fMRI but are important for brain function. Another example of the possible dangers of this argument is reported in my comments about the temporal domain. I think that the authors should carefully reconsider how they write these statements.

Subsection “Sub-voxel Smoothness” paragraph four: the problems related to temporal domain seem to be conceptualized in a way that is at odds with what we know of how fMRI is sensitive to the timing of neural population activity. The authors seem to put forward the idea that BOLD fMRI roughly corresponds to a firing rate averaged over long time windows, and that it will be insensitive to timing of spikes for example synchronous firing:

"BOLD will fail to measure other temporal coding schemes, such as neural coding chemes that rely on the precise timing of neural events, as required by accounts that posit that the synchronous firing of neurons is relevant to coding (Abeles, Bergman, Margalit, & Vaadia, 1993; Gray & Singer, 1989). Unless synchronous firing is accompanied by changes in activity that fMRI can measure, such as mean population activity, it will be invisible to fMRI".. "Because the BOLD signal roughly summates through time.."

This reasoning appears at odds with what concurrent recordings of neural activity and fMRI show. First, as the pioneering work of Logothetis et al. (Nature 2001) already revealed and many studies from his groups confirmed over the years, the BOLD correlates strongly with LFPs (a measure of mass synaptic activity) and it correlates with spike rates only when those correlate with LFPs. Second, the degree of millisecond-scale synchronization among neurons is not only picked by BOLD: it actually seems to be a primary correlate of fMRI BOLD, and much more so than the firing rate or multi-unit activity computed over long windows. One study of Logothetis group (Magri et al. J Neurosci 2012) showed that the primary correlate of BOLD is the γ-band LFP power. Γ band power expressed the strength of local neural synchronization over a scales of few ms to one or two tens of ms. These results are also reported in human studies using EEG with fMRI (see Scheeringa et al. Neuron 2011).

So, my suggestion is to rewrite completely the "temporal dimension" part of this paper. This can also serve as an example suggestion of how very dangerous it would be to rule out a coding scheme based considering the success of fMRI and its spatio-temporal limitations (see my comments above).

Reviewer #3:

Summary:

The authors applied an fMRI data analysis method called representational similarity analysis (RSA) to artificial neural network data. They argued that neural code must be both sub-voxel smooth and functionally smooth for RSA to recover the neural similarities from fMRI data.

Comments:

1) What is the definition of the term "functional smoothness"? In the "Functional smoothness" section, the authors only stated that factorial design coding and hash function coding are not functionally smooth, but neural network models are functional smooth. I only see examples but no definition.

2) If the main contribution of the paper is that the neural code must be smooth for RSA method to decode. Then the authors should provide necessity and sufficiency proofs of this statement (Discussion section first paragraph): (1) if RSA can decode the similarity in the fMRI data, then the neural code must be sub- voxel smooth and functional smooth. (2) As long as the neural code is sub- voxel smooth and functional smooth, RSA can encode the similarity in the fMRI data.

3) The authors should explain the reason they choose Deep Neural Network for their experiments. Friston 2003's dynamic causal model is a popular model for fMRI data simulation. Spiking neural network is another candidate used to study neural code. Please explain why Deep Neural Network is favorable for the experiments in this paper.

4) Experimental detail is lacking. There is no methods section. I also tried to look at the code the author provided at http://osf.io/v8baz, but the access was forbidden. It seems like the code folder is private. So there is not much I can comment on the methods used in this paper.

---

## [Author Response]

*Essential revisions:*

The three reviewers and I all agree that this paper raises an interesting set of questions. However, there were a number of concerns about the paper as it is currently written. Here are the main concerns as I see them, and how they should be addressed:

*– Two reviewers raise the question of whether your claims are fully supported by the results, suggesting in many cases that you are making strong claims that are not fully supported by your results. I think that in general the revision needs to better calibrate the claims to the evidence, and avoid overreaching. In particular, the fact that fMRI is successful does not mean that it is capturing all, or even the majority, of interesting neural signals. This issue needs to be clearly addressed in the revision.*

As detailed below, we made an effort throughout the manuscript to qualify our claims. The changes, following the reviewers’ suggestions, include being clear that the brain may use other coding schemes that are inconsistent with the success of fMRI. We make clear that our results do not rule out these coding schemes, but do suggest that the brain is using coding schemes that are to some extent consistent with fMRI’s success. To flesh out this idea, we introduce the notion that the neural code could consist of a mixture of coding schemes and that at least some portion of this mixture is consistent with the success of fMRI.

*– The concept of "functional smoothness" is central to your manuscript, but the reviewers raised multiple questions regarding the clarity of this concept in your paper. The revision needs to be much clearer about what this concept means and how it relates more specifically to the other concepts of smoothness.*

As detailed below, we made substantial improvements that include formalizing functional smoothness as a straightforward equation and applying this measure to the simulation results. We also made an effort to improve the discussion of functional smoothness, as well as to disentangle it from related concepts.

*– Reviewer 2 points out that your paper needs to make better contact with the literature that has investigated the neural basis of the BOLD signal. In particular, specific attention must be paid to the points raised by reviewer 2 regarding the implications for temporal aspects of the BOLD signal.*

We have incorporated all the literature that reviewer 2 suggested. In particular, reviewer 2’s pointers helped us improve the subsection on the temporal aspects of the BOLD signal.

*– Reviewer 3 points out that the paper lacks methodological details, and that they were not able to access the materials linked at OSF. In the revision, please include a detailed description of your methods along with a working link to the code and materials.*

We regret that the OSF repository was set to be private at the time of submission. We have made the repository public, which contains all scripts and readme files to guide the user. We also now provide complete methods to the simulations in the main text.

*I am including the reviewers' comments below, which provide greater detail regarding the issues outlined above.*

*Reviewer #1:*

[…]

*1) What are the scopes of the claims being made?*

*One main issue with the paper is the precise nature of the central claim. This claim varies in scope throughout the paper from more modest and well-evidenced to more all-encompassing and arguably beyond the evidence put forth. The main claim can loosely be phrased as: The fact that fMRI writ large “works” tells us something about the underlying neural mechanisms. The precise cashing out of this claim includes the following descriptions:*

Abstract “certain neural coding schemes are more likely than others” (Plausible but not necessarily true).

*Subsection “Sub-voxel Smoothness”, “the rate of change (i.e., smoothness) of neural activity must not exceed what can be measured within a voxel.” (This phrasing seems overly general – There is no reason why rates of change couldn't also exceed what can be measured. All that is compellingly shown is that a non-trivial part of the neural code behaves in a temporally smooth manner).*

Thank you, we now make clear that multiple codes could coexist and that our results imply that the mixture will contain components with certain properties.

In the same section “temporally demanding coding schemes cannot carry the day” (Not clear what “carry the day” means – If it means “Can't fully explain all patterns” then it is likely true and well supported, if it means 'temporally demanding coding schemes cannot play an important role also' I think it's too strong a claim).

*Also in this section “the neural code must be spatially and temporally smooth with respect to neural activity” (Overly general in that it implies there is only one neural code ('The')).*

Edited.

*Discussion section first paragraph “the neural code must be smooth, both at the sub-voxel and functional levels.” (overly general – It can be smooth and non-smooth simultaneously to currently unknown degrees, e.g. Figure 1 left and right could be overlapping).*

We revised Figure 1 and the supporting discussion to make the concepts clearer.

*Discussion section paragraph ten “However, fMRI's success might mean that when one is interested in the nature of computations carried out by the brain, the level of analysis where fMRI applies should be preferred.” (A considerable overgeneralisation – All we know for sure is that fMRI can provide important information, but there is no reason why lower (or higher) levels of temporal or spatial abstraction can't be even more informative.).*

There were already qualifications in surrounding passages, but we have further tempered this statement.

*More limited and arguably more accurate are:*

*– Abstract section “Deep neural network approaches, are consistent with the success of fMRI” (True and well-supported).*

*– Subsection “Sub-voxel Smoothness”, “The success of fMRI does not imply that the brain does not utilize precise timing information” (an important explication of the point raised above that multiple coding schemes could operate concurrently, and that fMRI would “work” if it only picks up parts).*

We agree and have increased discussion of what fMRI may be missing and the possibility of code mixtures.

*Similarly in subsection “Factorial Design Coding”, discussing factorial coding, the authors state “Interestingly, if any region in the brain had such a distribution of voxels, neural similarity would be impossible to recover by fMRI.” And later “If the neural code for a region was employing a technique similar to factorial design, neuroimaging studies would never recover similarity structures by looking at the patterns of active voxels in that region.” These statements are only true if other coding schemes aren't allowed to co-exist. It may well be that factorial coding schemes co-exist alongside spatially smooth codes but simply cannot be detected using current methods and/or analyses. Or, to couch it in terms of Figure 1, there is no reason why the neural code might not be an overlay, or mixture, of A and B. This implicit suggestion that there must be a single coding scheme is reflected in the title, which refers to “The” neural code.*

This is addressed in the mixture concept, as well as qualifying statements throughout to make clear such results would hold only if that were the “sole” code used. One contribution of this paper is to work through the properties of several coding schemes with respect to the BOLD response. In some circumstances, it is helpful to consider the properties of single coding schemes in isolation for purposes of clarity, though we make clear that the brain may utilize mixtures.

*As can be seen when read in succession, these are not interchangeable epistemic claims – Some are “merely” existence claims whereas others rule out alternative explanations.*

*In our view, the main claim of that article, that is important and well supported, can be phrased as follows:*

The success of fMRI makes it highly likely that at least some non-trivial subset of the signal/computations made by the neural code must be smooth.

*Or, in the authors words “One general conclusion is that important aspects of the neural code are spatially and temporally smooth at the sub-voxel level”.*

*In other words, the article provides solid evidence for a more modest positive claim but at times shifts into language that suggests it has ruled out (or rendered exceedingly unlikely) other neural codes. We do not believe this is the case (nor, for that matter, do I think this is an achievable goal in the context of any single paper, for both principled mathematical reasons that every data pattern is compatible with an infinity of data generating mechanisms, and the pragmatic reasons of scope). In other words, we would suggest that the precise scope of the central claim is delineated more clearly, and that this scope is retained throughout*

We believe that the claims and the supporting evidence are now better aligned.

*2) Is "sub-voxel smoothness" an accurate and clear description of what is required for successful fMRI?*

*The other main challenge in the article is the introduction of two new terms, spatial and functional smoothness. It seems as though these are not entirely precisely defined, and to the extent that they are, one might wonder whether a) “smoothness” is the best term and b) what, if anything, is similar about spatial and functional smoothness such that they share a conceptual term. We outline these challenges below.*

*With regards to spatial smoothness, I wonder whether the term inhomogeneity isn't more appropriate or central to the claim being made. For instance, it seems as though one could rearrange the voxel columns in Figure 1 whilst preserving sub-voxel smoothness as well as functional smoothness – The arrangement of the voxels as a gradient could be misunderstood as being about super-voxel smoothness (especially in the light of evidence for early visual retinotopic mapping). (more detail below).*

We are essentially in agreement, though perhaps this was not clear in the original submission. Indeed, we retitled a subsection “Voxel Inhomogeneity Across Space and Time” and have used this language throughout in light of your comments. Subvoxel smoothness in the original manuscript was motivated by the mathematical concept of smoothness, which involves “clumpiness” and lack of sharp discontinuities. In other words, the concept does not imply smooth global changes in responses and would be consistent with your thought experiment of rearranging elements. One peril in related thought experiments is assuming that the brain somehow “knows” the voxel size or that voxels favorably align.

*The paper distinguishes between the concepts of sub-voxel and super-voxel smoothness. These concepts appear to be independent – i.e. one could have subvoxel smoothness combined with non-smooth super-voxel patterns. i.e., it's not entirely clear why “In such a spatially smooth representation, the transitions from red to yellow occur in progressive increments” follows necessarily from subvoxel smoothness? The later definition that voxel size can be changed arbitrarily does capture this, but for a *given* voxel size a pattern could be subvoxel smooth and super-voxel non-smooth. Note that the empirical examples outlines in subsection “Sub-voxel Smoothness” are compelling evidence that sub and super-voxel smoothness are related, but in principle they needn't be (this point is made by the authors in subsection “Functional Smoothness” with respect to functional smoothness).*

*[…]*

*Picturing these four alternatives, it seems that in order for different stimuli to create different patterns at the level of fMRI voxels, the important factor is neither sub-voxel nor super-voxel smoothness, but something which might be termed "across-voxel inhomogeneity" – i.e. that different voxels sample populations of neurons with different functional properties. For example, in the example just suggested, wherein the “columns” of the left-hand panel are shuffled, there would no longer be a smooth gradient in the functional selectivities of voxels as one moves across the cortical sheet, but different stimuli could still evoke unique multi-voxel patterns.*

We now better define functional smoothness, going as far as to formalize it. We made an effort in the original manuscript to note how functional and super-voxel smoothness were distinct, which is what motivated the inclusion of Figure 4 (now Figure 3). We now go further throughout and have added a paragraph discussing how these concepts diverge (subsection “Functional Smoothness”).

We also discussed the importance of voxel size in the original manuscript, but these points were too obscured. We now highlight these points more (subsection “Voxel Inhomogeneity Across Space and Time”) and have amended the Figure 1 example to demonstrate the role of voxel size. This might be a case of agreement that was not made sufficiently clear in the original manuscript.

*It is important here to engage with previous literature on fMRI decoding, especially Kamitani & Tong (2005) and subsequent reflections on why stimulus orientation can be decoded from V1 via fMRI (e.g. Alink et al. 2013; Carlson, 2014). Orientation columns in V1 are arranged in a similar fashion to the "sub-voxel non-smooth" depiction in Figure 1, yet orientation can reliably be decoded. One plausible reason is that the chance of each voxel sampling all orientation selectivities exactly equally is very low, so even if neuronal selectivities are intermixed at a sub-voxel level, slight differences in orientation preference are likely to emerge at the voxel level.*

We enjoyed reading these papers and now cite Kamitani and Tong (2005) and Alink et al. (2013) as additional examples of successful decoding.

*The caption of Figure 1 suggests non-smoothness is preserved “regardless of the precise boundaries of the voxels which quantize the brain” but it seems that this would only be true conditional on preserving approximate voxel size? If I am allowed to change both the size and boundaries I could create mostly-yellow and mostly-red voxels from Figure 1? Moreover, as mentioned above, random variations in sampling can lead to successful decoding of schemes similar to the non-smooth pattern.*

Issues of voxel size are covered above.

*Sub-voxel smoothness might not be the most intuitive term to use, not only for the reasons mentioned under point 2, but also because it implies spatial and not temporal smoothness. It seems more appropriate to talk about spatial and temporal scales at which useful information (about stimuli, thoughts, actions) is represented by brain activity.*

This issue has also been covered above.

*3) "Functional smoothness" is a relational property between two models or between data and a model, not an inherent property of one model.*

*Arguably the most challenging term introduced by the paper is "functional smoothness." It is not made clear what this is a property of. From the definition, that "similar stimuli map to similar representations," a model is "functionally smooth" with respect to some second representational model if it preserves the representational geometry of that target model. In the first set of simulations involving noise-corrupted images, the target model is pixel space. In the second set of simulations involving objects of different categories being presented to a deep neural network, the target model is a semantic or conceptual space. In other words, functional smoothness is not an objective property of the model, but a relational property between an input and an output space. This seems to be recognised in some sections, e.g. subsection “Vector Space Coding”: "vector space coding is functionally smooth in the trivial sense as the function is identity."*

We now make the definition of functional smoothness more clear and discuss after the definition how it can apply.

We also removed the random network simulations involving images and instead report the simulations with the Gaussian vectors. We only included the simulations with the images because we thought they would be more intuitive, but that seems to not be the case and they might lead to confusion in some readers with the deep learning simulations that classify photographs.

*However, many other phrases in the paper imply that smoothness is intrinsic to models, e.g.:*

*– subsection “Deep Learning Networks”: "one key question is whether functional smoothness breaks down at more advanced layers in DLNs…"*

*– subsection “Deep Learning Network Simulation”: "We consider whether functional smoothness declines as stimuli progress to more advanced layers…"*

*– and: "Violating functional smoothness, all other similarity structure is discarded…"*

These issues are addressed in a comment above.

*The simulations underlying Figure 2 and Figure 3 add to this confusion rather than illustrating functional smoothness. Presumably the weights within the weight matrices used in the models from "matrix multiplication coding" onwards were random? If so, then Figure 3 seems to illustrate the trivial effect that as one performs increasingly many random nonlinear transformations on an input, distances between stimuli in the input space will be less and less well correlated with distances in the output. This result doesn't seem to reveal anything fundamental about the merit of (non-random) deep nonlinear networks as models of brain representation.*

We made the methods for these simulations more clear. Yes, the random network simulations show that functional smoothness will break down, even in an untrained network, such that more advanced layers should be harder to recover similarity spaces from using fMRI. Prior to this paper, I don’t think people appreciated the basic geometry of projecting vectors repeatedly would have these effects as every neural network paper we have read focuses on the role of training! Correspondingly, imaging researchers speak of the signal to noise ratio or noise ceiling of various regions as if these are basic properties of those regions as opposed to a consequence of receiving input that has been several times over quasi-linearly transformed.

*To highlight the fact that functional smoothness is a relational property, it might instead be helpful to create a simulation akin to these, but using a trained network, and adding noise in a more abstract "target space" (e.g. noise could be added to the location of an object within an image, for example by sliding around the location of a dog image superimposed on a natural background). The earliest layer of the network should be "functionally smooth" w.r.t. changes in pixels but not location (i.e. pixelwise differences will make a large difference to early representations); middle layers may be "functionally smooth" w.r.t. location but not pixels (e.g. the same dog at nearby locations will activate the same feature detectors looking for eyes, legs, etc), while the final categorical layer would ideally be wholly invariant both to pixel and location changes provided the image continues to contain a dog.*

I think the simulations should now be clearer as we have replaced the dog and truck image inputs with vectors drawn from a Gaussian.

*It would also help to distinguish between coding schemes that have the capacity to be functionally smooth, vs. those that actually functionally smooth, with respect to a particular input space. Factorial and hash coding are raised as examples of codes that are "not functionally smooth," while neural-network-style encodings of various complexity are evaluated for whether the simulated examples happen to be functionally smooth with respect to the input spaces (pixel space, then semantic similarity space). However, there is an important difference in the sense in which factorial coding is "not functionally smooth" and that in which the late layers of GoogLeNet are not – factorial and hash coding are not capable of being functionally smooth (because every representation is orthogonal to every other), whereas the neural networks considered are capable in principle of strongly preserving the representational geometry of any input space desired (although see Point 6).*

Interestingly, the neural networks essentially do hash coding as additional layers are added. We note this connection in two places in the manuscript, final paragraph of subsection “Hash Function Coding” and Discussion section paragraph three. Like the neural networks, intermediate steps in a hash coding algorithm may be functionally smooth with only the end step non-smooth,

*4) What is the relationship, if any, between spatial and functional smoothness?*

*It seems as if spatial and functional smoothness, as defined in the paper, should be completely independent of one another, and they are described in paragraph two of subsection “Functional Smoothness” as "distinct concepts." However:*

*Subsection “Matrix Multiplication Coding”: "…the internal representation of this coding scheme is completely nonsensical to the human eye and is not super-voxel smooth."*

*This is confusing for two reasons. First, there is no reason for any internal representation to be "sensible to the human eye." The fact that the input representation is "sensible" is just an artefact of using images as the inputs. Even then, the image is only understandable by the eye if one preserves the exact spatial order of units and arranges them into rows and columns of exactly the right dimensions. Any shuffling or re-arrangement of the pixels would constitute an identical representation, and yet would no longer be sensible to the eye – so sensibility to the eye does not seem relevant.*

You are grasping our intended point perfectly – that figure and caption were included to demonstrate a case in which functional smoothness is preserved but super-voxel smoothness is not. We now make this clearer, as well as expand on the differences between these concepts. Particular care was paid to this issue in the section on Voxel Inhomogeneity Across Space and Time and in the caption of what is now Figure 3.

Second, how does this minimal model (a transformation of the input by a single matrix multiplication) specify properties about the spatial arrangement of voxels? Paragraph two subsection “Functional Smoothness” states that functional smoothness is defined at the level of voxels (although this seems a little counter-intuitive, since brains and neural networks encode information at the level of neurons). So in the "matrix multiplication code," we should imagine a case where there are as many voxels in the output as there are pixels in the input image, and the activation level of each one is determined by an arbitrary linear combination of all input pixels. This output will be "functionally smooth" w.r.t. pixel space if the matrix transformation is one that preserves representational geometry (e.g. a rotation matrix; see first comment under "Smaller points" below for more on this…). This will be true however one arranges the voxels spatially. Some possible arrangements will appear super-voxel smooth (e.g. if voxels are placed next to those with the most similar selectivities), and some will not (e.g. if voxels are randomly placed), but all arrangements will be functionally smooth.

We moved away from the images in the random networks to reduce this confusion.

*If there is some deeper connection between functional and spatial smoothness, this needs to be more clearly explained and illustrated.*

Thank you, as mentioned above this was one of the major revisions of the paper. We also view it as critical to make this distinction clear.

*5) Different neural code properties may be required for "successful fMRI" when doing mean-activation vs. decoding vs. representational similarity analyses.*

*The title-bearing central claim refers to the “success” of brain imaging, but it is not entirely clear how this should be conceived. It might be worth briefly describing what counts as success, and how each result constrains possible neural codes. For example, it would be good to separately consider what findings from (1) old school mean-activation “blobology”, (2) classifiers performing above chance, (3) RSA imply about neural coding. It seems uncontroversial that all require "temporal smoothness" and some form of across-voxel spatial inhomogeneity in order for different stimuli to create detectably different fMRI activations. But they may have different further implications, for example:*

*1) The success of "blobology" in finding multi-voxel clusters with similar functional properties suggests some degree of super-voxel smoothness.*

*2) Above-chance decoding does not seem to even require a neural code capable of functional smoothness. E.g. in a factorial code, although every stimulus elicits a pattern orthogonal to that elicited by any other, one could still do successful "mind-reading" as long as one had access to data from previous trials on which subjects had viewed that stimulus.*

*3) The success of RSA (i.e. finding interesting and nuanced similarity patterns between patterns evoked by different stimuli, which seem to bear some relation to the geometry of those stimuli within other models such as pixel space or semantic space and can be compared to predictions from computational models) does require a neural code which is capable of functional smoothness. The importance of functional smoothness only to RSA does seem to be recognised in the paper, but could be made more explicit, e.g.:*

These are good points which we tried to bring forward throughout and in particular in an added paragraph in the Discussion, paragraph seven.

*Subsection “Factorial Design Coding” paragraph five: "If the neural code for a region was employing a technique similar to factorial design, neuroimaging studies would never recover similarity structures by looking at the patterns of active voxels in that region."*

*One thing to note here is that model comparisons do not necessarily require RSA. Does the success of other analysis methods that compare models (e.g. voxel-receptive-field modelling) also point to functional smoothness? Or is this term strongly linked to RSA as an analysis framework, to the extent that it does not have a meaning outside it? If so, why do the authors focus so strongly on functional smoothness? (RSA is successful, but so are (linear) classifiers).*

We focused on recovering similarity spaces, but do discuss the implications for other analysis approaches in the aforementioned added Discussion paragraph.

*6) Simulations with a network trained on a task other than categorisation would help justify the claim that "non-smoothness" is an inevitable property of deep nonlinear neural networks.*

*The final simulations, in which distances within a “semantic space” are informally compared to distances within successive layers of the deep neural network GoogLeNet, conclude that later layers are less functionally smooth w.r.t. semantic space than early ones (since they lose between-category similarity information), and that “the decline in functional smoothness at later layers does not appear to be a straightforward consequence of training these networks to classify stimuli.” This latter conclusion is likely to be controversial, and is not strongly supported by the sparsity analysis.*

The simplest way to clarify the contribution of the training task to the representational geometry in the final layers would be to show RDMs from (a) randomly-weighted networks, and (b) an unsupervised network, or one trained on a task orthogonal to categorisation. A good candidate would be the unsupervised seven layer neural network in Wang & Gupta (2015), which is available from https://github.com/xiaolonw/caffe-video_triplet

*Again though, this would not reveal anything about the "functional smoothness of the model," since that is not an inherent property, but only the similarity between the representations within the model and in semantic space.*

This was a very nice suggestion. We spent some weeks getting their code to run and made some interesting discoveries. In this literature, classification decisions are made by unsupervised networks by first training the networks on unlabelled data (i.e., unsupervised learning). The final layer of such networks is then considered to have distilled the meaning of the image. To evaluate the performance of the unsupervised network, a simple classifier is then trained using this final layer as input. Wang and Gupta did not include this critical component in their code repository, but we corresponded with them extensively on email to discuss the myriad of ways in which they boosted performance with their network.

Unfortunately, the bottom line is that their unsupervised network coupled with a classifier operating over the final layer only retrieves the correct label for an image in its top 20 guesses 40% of the time, which is not in the same league as InceptionV3 model which achieves 97% accuracy in only its top 5 guesses. In short, we can’t use this mostly unsupervised network to compare to the supervised network because it can’t properly classify images. In light of their model’s troubles, we made an edit to our discussion unsupervised methods that now focuses on performance rather than scope.

*7) Previous work*

*One key feature missing from this paper is closer engagement with previous literature on decoding, such as the seminal findings in Kamitani & Tong (2005) (discussed above in Point 2), computational accounts (e.g. Kriegeskorte, N., Cusack, R., & Bandettini, P. (2010) or de Beeck, H. P. O. (2010), and those discussing the plausibility of more trivial structural explanations such as differences in vasculature (e.g. Shmuel, A., Chaimow, D., Raddatz, G., Ugurbil, K., & Yacoub, E. (2010)).*

We included some of this work in places where we could find a fit.

*Miscellaneous*

*Subsection “Matrix Multiplication Coding” states “Matrix multiplication maps similar inputs to similar internal representations.” The claim seems to refer to multiplication of a 1xn input by an arbitrary nxn matrix (such as would be implemented by a 1-layer fully connected linear neural network). Although there are some such matrix multiplications which would perfectly preserve distances between different inputs in their original vs. transformed spaces (e.g. rotation matrices), and with random matrices, distances in the original and transformed spaces will tend to correlate (as your simulations show), the claim is not generally true. There are many matrix multiplications which will completely disrupt representational geometry.*

The methods now make clear that the matrices were random, which makes it extremely unlikely they will be rank deficient. As a further safeguard, the simulation results are averaged over 100 different networks, which is now made clear in the methods description.

*Subsection “Deep Learning Network Simulation” paragraph three states that sparseness of representation does not decline for a "later advanced layer" of the category-supervised deep neural net. Which layer is this? It seems surprising that sparseness does not increase in at least the final layer (i.e. the output of the softmax operation). Relatedly, is it worth showing more layers in Figure 5? If not, why are these two layers shown?*

We tried to keep it simple and ensure that the comparison was not biased in our favor. We chose these layers because sparsity (which we quantified with the Gini coefficient) goes against our hypothesis. We also tried to compare two layers that had the same structure (both pooling layers) to make sure we were comparing apples to apples. We didn’t explicitly number the layers because the architecture of the model is so complicated that three people may number the layers three different ways. Fortunately, the OSF repository and code makes clear which structures we used from InceptionV3 for those interested.

*Related to Point 3 above, it would help to use more precise language about the nature of the sensory inputs or brain representations being discussed. For example, subsection “Matrix Multiplication Coding” says that a particular coding scheme "takes an input stimulus (e.g. a dog) and multiplies it by a weight matrix" – given that a dog is not the sort of thing that can be multiplied by a weight matrix, does this mean either (specifically) an image of a dog, or (generally) the activity within a preceding layer of neurons in a neural network model? Referring to the (arbitrary) input images in the simulations as "prototypes" is also confusing, as it suggests they have some special status to the models.*

Fixed. Again, we switched to Gaussian vectors here to avoid such confusions. We do retain the one figure with the image matrix multiplication because it makes it easier to see that even when the network states look “scrambled” they actually preserve functional smoothness, which was discussed above.

*– Although I like the hash coding example in subsection “Hash Function Coding” says it's “potentially useful for biological systems” – it might be worth elaborating briefly in what circumstances a biological system would evolve something akin to hashcoding for certain stimuli? It seems rather inefficient and hard to reconcile with basic facts about learning (e.g. co-occurrence increasing association strengths, and thereby similarity) and memory (e.g. semantic co-activation)? I appreciate the later evidence for the compatibility of higher layers with hash coding but the above claim seems more general – This relates to the above discussion on what.*

We mention one possible mental function in the final paragraph of subsection “Hash Function Coding”.

*– The selection of coding schemes cover interesting (and rhetorically convincing) ground but the motivation for this set of coding schemes doesn't seem to be motivated. E.g., why focus on hash coding and factorial mapping – Are those a subset of a broader range that could be considered? Something similar could be said about the selection of NN algorithms. The selection of codes seems to constitute an input being transformed by successively more complex neural networks ("vector space coding" = no transformation of the input; "gain control coding" = one nonlinearity; "matrix multiplication coding" = one linear transformation; "perceptron coding" = one linear transformation plus one nonlinearity; "multi-layer neural network"….etc). Although this is logical, the descriptions (e.g. "vector space coding") misleadingly imply that these are qualitatively distinct strategies for encoding a stimulus, and leave it to the reader to discover the logic of selecting these particular "codes".*

We chose popular coding schemes and models that helped illustrate the key concepts. We state in subsection “Vector Space Coding” that the neural network models are all components that can be assembled to form one larger model, as opposed to distinct coding strategies. Hopefully, this nicely transitions to the deep learning network simulation.

*– The paper by Bracci & de Boeck (2016) seems worth discussing in more detail, as it provides potential direct evidence for the hierarchy of smoothness. One wonders whether there are plausible alternative explanations that should be taken into account wrt varying levels of prediction accuracy across the ventral stream? For instance, the noise ceiling also often goes down (i.e. there is less signal to be explained in principle).*

We agree and actually mention this work both in the original submission and revision (Discussion section paragraph six). We suggest that our work may help explain this noise ceiling.

*Reviewer #2:*

*This paper addresses an interesting subject, that of what we can learn about the neural code from fMRI. The paper makes a valuable conceptual effort to think about which neural codes are supported by fMRI observations and which are not. Much as I like the paper and I think it is important to discuss these issues, I think that the connection between neural activity and fMRI, which should be central to this topic, is not sufficiently well discussed. My fear is that places of the current manuscript would look insufficiently developed to neurophysiologists investigating neural coding. In the following I raise the attention of the authors to what are in my view problems in the current manuscript that need addressing, and I provide a few suggestions. I hope that this will improve their paper.*

*The current writing of the paper may be taken at specific places to argue that the success of fMRI implies that a coding scheme that does not come though fMRI is not one used by the brain to compute. ("Through proof and simulation, we determine which coding schemes are plausible given both fMRI's successes and its limitations in measuring neural activity"… "The neural code must have certain properties for this state of affairs to hold. What kinds of models or computations are consistent with the success of fMRI?" … "The success of fMRI does not imply that the brain does not utilize precise timing information, but it does mean that such temporally demanding coding schemes cannot carry the day given the successes fMRI has enjoyed in understanding neural representations.").*

As mentioned in response to reviewer 1, we have made efforts to moderate these claims.

*Of course there would be no basis for such a strong claim, and the authors should state and discuss this clearly. It is for example possible that fMRI gets only a part of the neural code used by the brain, and that other parts, perhaps as important as others, are simply lost by the limitations of fMRI but are important for brain function. Another example of the possible dangers of this argument is reported in my comments about the temporal domain. I think that the authors should carefully reconsider how they write these statements.*

Thank you. We have done so and made an effort throughout to correct the tone and try to find the balancing point between claims and supporting evidence.

*Subsection “Sub-voxel Smoothness” paragraph four: the problems related to temporal domain seem to be conceptualized in a way that is at odds with what we know of how fMRI is sensitive to the timing of neural population activity. The authors seem to put forward the idea that BOLD fMRI roughly corresponds to a firing rate averaged over long time windows, and that it will be insensitive to timing of spikes for example synchronous firing:*

*"BOLD will fail to measure other temporal coding schemes, such as neural coding chemes that rely on the precise timing of neural events, as required by accounts that posit that the synchronous firing of neurons is relevant to coding (Abeles, Bergman, Margalit, & Vaadia, 1993; Gray & Singer, 1989). Unless synchronous firing is accompanied by changes in activity that fMRI can measure, such as mean population activity, it will be invisible to fMRI".. "Because the BOLD signal roughly summates through time.."*

This was not our intent and these passages have been heavily edited in response to reviewer 2’s comments. The overarching intended point was that BOLD will be blind to such temporal coding schemes unless they have a correlate that BOLD is sensitive to.

*This reasoning appears at odds with what concurrent recordings of neural activity and fMRI show. First, as the pioneering work of Logothetis et al. (Nature 2001) already revealed and many studies from his groups confirmed over the years, the BOLD correlates strongly with LFPs (a measure of mass synaptic activity) and it correlates with spike rates only when those correlate with LFPs. Second, the degree of millisecond-scale synchronization among neurons is not only picked by BOLD: it actually seems to be a primary correlate of fMRI BOLD, and much more so than the firing rate or multi-unit activity computed over long windows. One study of Logothetis group (Magri et al. J Neurosci 2012) showed that the primary correlate of BOLD is the γ-band LFP power. Γ band power expressed the strength of local neural synchronization over a scales of few ms to one or two tens of ms. These results are also reported in human studies using EEG with fMRI (see Scheeringa et al. Neuron 2011).*

We have reworked the discussion to center upon these contributions. One point that we enjoyed in Magri et al. (2012) that is now a center point of this section is that different mixes of activity at different bands can lead to an identical BOLD response. These kinds of points are exactly what we tried to bring out in this section in the original manuscript. We now believe we are more successful in doing so.

*So, my suggestion is to rewrite completely the "temporal dimension" part of this paper. This can also serve as an example suggestion of how very dangerous it would be to rule out a coding scheme based considering the success of fMRI and its spatio-temporal limitations (see my comments above).*

That was always part of our intended message, which should now be clearer. At a very broad level, we needed to bring out these issues clearly because establishing these views makes it surprising (i.e., informative) at some level that BOLD is useful in recovering similarity spaces, despite limitations in what it measures.

*Reviewer #3:*

*Summary:*

*The authors applied an fMRI data analysis method called representational similarity analysis (RSA) to artificial neural network data. They argued that neural code must be both sub-voxel smooth and functionally smooth for RSA to recover the neural similarities from fMRI data.*

Thank you for your time and useful input. As discussed below, we have improved the formal presentation in light of reviewer 3’s comments.

*Comments:*

*1) What is the definition of the term "functional smoothness"? In the "Functional smoothness" section, the authors only stated that factorial design coding and hash function coding are not functionally smooth, but neural network models are functional smooth. I only see examples but no definition.*

We tried to be clear, but at times a formal definition is useful. We now offer a formal definition (Equation 1) and apply this measure to the simulations. For example, the measure of functional smoothness does decline as layers in a neural network are traversed.

*2) If the main contribution of the paper is that the neural code must be smooth for RSA method to decode. Then the authors should provide necessity and sufficiency proofs of this statement (Discussion section first paragraph): (1) if RSA can decode the similarity in the fMRI data, then the neural code must be sub- voxel smooth and functional smooth. (2) As long as the neural code is sub- voxel smooth and functional smooth, RSA can encode the similarity in the fMRI data.*

We now make clear that similarity spaces by definition require functional smoothness (Equation 1). We discuss how RSA solutions and decoding solutions without functional smoothness would be degenerate in that any positive results would need to be driven by self-similarity.

*3) The authors should explain the reason they choose Deep Neural Network for their experiments. Friston 2003's dynamic causal model is a popular model for fMRI data simulation. Spiking neural network is another candidate used to study neural code. Please explain why Deep Neural Network is favorable for the experiments in this paper.*

We now explain the motivation for this choice on subsection “Deep Learning Networks”. Importantly, these models are not only simulating neural data, but are actually completing the behavioral task (in this case object recognition) at human-level proficiency.

*4) Experimental detail is lacking. There is no methods section. I also tried to look at the code the author provided at http://osf.io/v8baz, but the access was forbidden. It seems like the code folder is private. So there is not much I can comment on the methods used in this paper.*

We apologise for not making the repository public at the time of submission. It is now public with all code fully documented. We also have provided full methods for the simulations in the main text.